# Novel Affibody Molecules Specifically Bind to SARS-CoV-2 Spike Protein and Efficiently Neutralize Delta and Omicron Variants

Wangqi Du,[a] Peipei Jiang,[a,b] Qingfeng Li,[a] He Wen,[a] Maolin Zheng,[a] Jing Zhang,[a] Yanru Guo,[a] Jia Yang,[a] Weixu Feng,[a] Sisi Ye,[b] Saidu Kamara,[a] Pengfei Jiang,[a] Jun Chen,[a] Wenshu Li,[a] Shanli Zhu,[a] Lifang Zhang[a]

[a]Institute of Molecular Virology and Immunology, Department of Microbiology and Immunology, School of Basic Medical Sciences, Wenzhou Medical University, Wenzhou, Zhejiang, China
[b]The First Affiliated Hospital of Wenzhou Medical University, Wenzhou, Zhejiang, China

**ABSTRACT** The severe acute respiratory syndrome coronavirus 2 (SARS-CoV-2) pandemic has been an unprecedented public health disaster in human history, and its spike (S) protein is the major target for vaccines and antiviral drug development. Although widespread vaccination has been well established, the viral gene is prone to rapid mutation, resulting in multiple global spread waves. Therefore, specific antivirals are needed urgently, especially those against variants. In this study, the domain of the receptor binding motif (RBM) and fusion peptide (FP) (amino acids [aa] 436 to 829; denoted RBMFP) of the SARS-CoV-2 S protein was expressed as a recombinant RBMFP protein in *Escherichia coli* and identified as being immunogenic and antigenically active. Then, the RBMFP proteins were used for phage display to screen the novel affibody. After prokaryotic expression and selection, four novel affibody molecules (Z14, Z149, Z171, and Z327) were obtained. Through surface plasmon resonance (SPR) and pseudovirus neutralization assay, we showed that affibody molecules specifically bind to the RBMFP protein with high affinity and neutralize against SARS-CoV-2 pseudovirus infection. Especially, Z14 and Z171 displayed strong neutralizing activities against Delta and Omicron variants. Molecular docking predicted that affibody molecule interaction sites with RBM overlapped with ACE2. Thus, the novel affibody molecules could be further developed as specific neutralization agents against SARS-CoV-2 variants.

**IMPORTANCE** SARS-CoV-2 and its variants are threatening the whole world. Although a full dose of vaccine injection showed great preventive effects and monoclonal antibody reagents have also been used for a specific treatment, the global pandemic persists. So, developing new vaccines and specific agents are needed urgently. In this work, we expressed the recombinant RBMFP protein as an antigen, identified its antigenicity, and used it as an antigen for affibody phage-display selection. After the prokaryotic expression, the specific affibody molecules were obtained and tested for pseudovirus neutralization. Results showed that the serum antibody induced by RBMFP neutralized Omicron variants. The screened affibody molecules specifically bound the RBMFP of SARS-CoV-2 with high affinity and neutralized the Delta and Omicron pseudovirus *in vitro*. So, the RBMFP induced serum provides neutralizing effects against pseudovirus *in vitro*, and the affibodies have the potential to be developed into specific prophylactic agents for SARS-CoV-2 and its variants.

**KEYWORDS** severe acute respiratory syndrome coronavirus 2, SARS-CoV-2, S protein, affibody, neutralization, SPR, receptor binding motif, RBM, fusion peptide, FP

Address correspondence to Shanli Zhu, wenzhouzsl@126.com, or Lifang Zhang, wenzhouzlf@126.com.

The authors declare no conflict of interest.

Severe acute respiratory syndrome coronavirus 2 (SARS-CoV-2) is the causative agent of coronavirus disease 2019 (COVID-19) (1), which has spread globally and threatens the public health system worldwide (2). Because of its highly contagious nature

and mortality rate (3), the virus caused more than 585 million people to be infected and 6.4 million dead by early August 2022 (https://covid19.who.int). The SARS-CoV-2 spike (S) protein is the major target for vaccines and antiviral drug development (4, 5), and its vaccines include the mRNA, adenoviral-vectored, protein subunit, and inactivated whole-virion vaccines (6). However, a reduction in neutralization titers using convalescent or vaccine sera has been discovered for Delta (7) and Omicron (8–10) variants and has led to some breakthrough infections. Therefore, new specific agents against variants are needed urgently.

SARS-CoV-2 is an enveloped virus with positive-sense single-stranded RNA (+ssRNA) of approximately 30 kb (11). The virus genome expresses four structural proteins, including Spike (S), envelope (E), membrane (M), and nucleocapsid (N) (12). The S protein, a trimer structural glycoprotein consisting of 1,273 amino acids, plays a key role in the viral infection process (13–15). During the process, the S protein was cleaved into S1 and S2 subunits at the S1/S2 cleavage site (amino acids [aa] 685/686) by furin protease, and the S1 subunit RBM domain (aa 437 to 508) was exposed to bind to ACE2 in host cells to initiate viral attachment (16). Then, the S2 subunit is cleaved into the S2' subunit (aa 816 to 1273) by transmembrane serine protease 2 (TMPRSS2) at the S2' cleavage site (aa 815/816), and the hidden fusion peptide (FP) domain (aa 816 to 826) was exposed and anchored into the cell membrane for membrane fusion (17, 18). After fusion, the virus releases its RNA into cells for replication (16) and then releases virions after intracellular assembly. In addition, researchers confirmed that the furin cleavage sites (FCSs; aa 681 to 685; aa sequence: PRRAR) of the S protein might facilitate the SARS-CoV-2 to be more contagious (19, 20), and numerous researchers have characterized that SARS-CoV-2 FCS deletion mutants significantly attenuated viral infection (21, 22). Accordingly, the RBM domains, FP domains, and cleavage sites of the S protein play key roles during the replication cycle of the SARS-CoV-2, involving adsorption, membrane fusion, and penetration into host cells.

Antivirals of novel monoclonal antibodies (MAbs), such as etesevimab and bamlanivimab based on the SARS-CoV-2 S protein, have already been developed (5) and authorized for emergency use against COVID-19. However, only a few of these candidates have been tested for their capability of neutralizing variants (23), and the emergence of new variants may even further undercut the therapeutic efficacy of MAbs (7, 24, 25). Furthermore, the antibodies are large, namely, 150 kDa in size; the S protein of SARS-CoV-2 is extensively glycosylated; and glycans shield about 40% of the protein surface of the S trimer (26), which could mask the recognition sites of S protein and reduce the neutralizing effects of antibodies against SARS-CoV-2 (27, 28). Thus, novel specific agents beyond MAbs against SARS-CoV-2 are urgently in need.

Affibody is a small molecular protein composed of 58 amino acids (6.5 kDa) consisting of 3-$\alpha$-helix structure (29), which is much smaller but retains the essential specificities and affinities of the antibody (30). With high affinity and specificity binding to target proteins (31), it can especially recognize certain antigens and can be used in diagnostic and therapeutic applications (32). Affibody molecules derive from the engineered Z-domain of *Staphylococcal* protein A (SPA-Z), based on the IgG-binding B-domain of *staphylococcal* protein A with mutation of the 29th amino acid glycine into alanine, and thus is more stable for pH changes and is highly soluble in aqueous solutions (33). In the IgG Fc-binding sites of the Z-domain located in the first and second $\alpha$-helix, 13 amino acids in this region can be randomized substitutions without altering their basic structure (34). Through molecular cloning technology to randomly mutate the 13 residues, a total of $20^{13}$ affibody molecules could be clustered in the primary affibody library, and they bind to any given protein theoretically (33, 35) (Fig. 1). More than 40 affibodies have been developed for potential targeted therapy or imaging diagnosis candidates. Among them, the affibodies could target the pathogenic proteins, including the HIV-1 envelope glycoprotein 120 (HIV-1 gp120) (36), Epstein-Barr virus latent membrane protein 2 (EBV-LMP) (37), human papillomavirus (HPV) type 16 and 18 E7 proteins (34, 38–40), *Chlamydia trachomatis* major outer membrane proteins (Ct-MOMP) (41), and HPV16 E6 protein (38). Moreover, the most promising

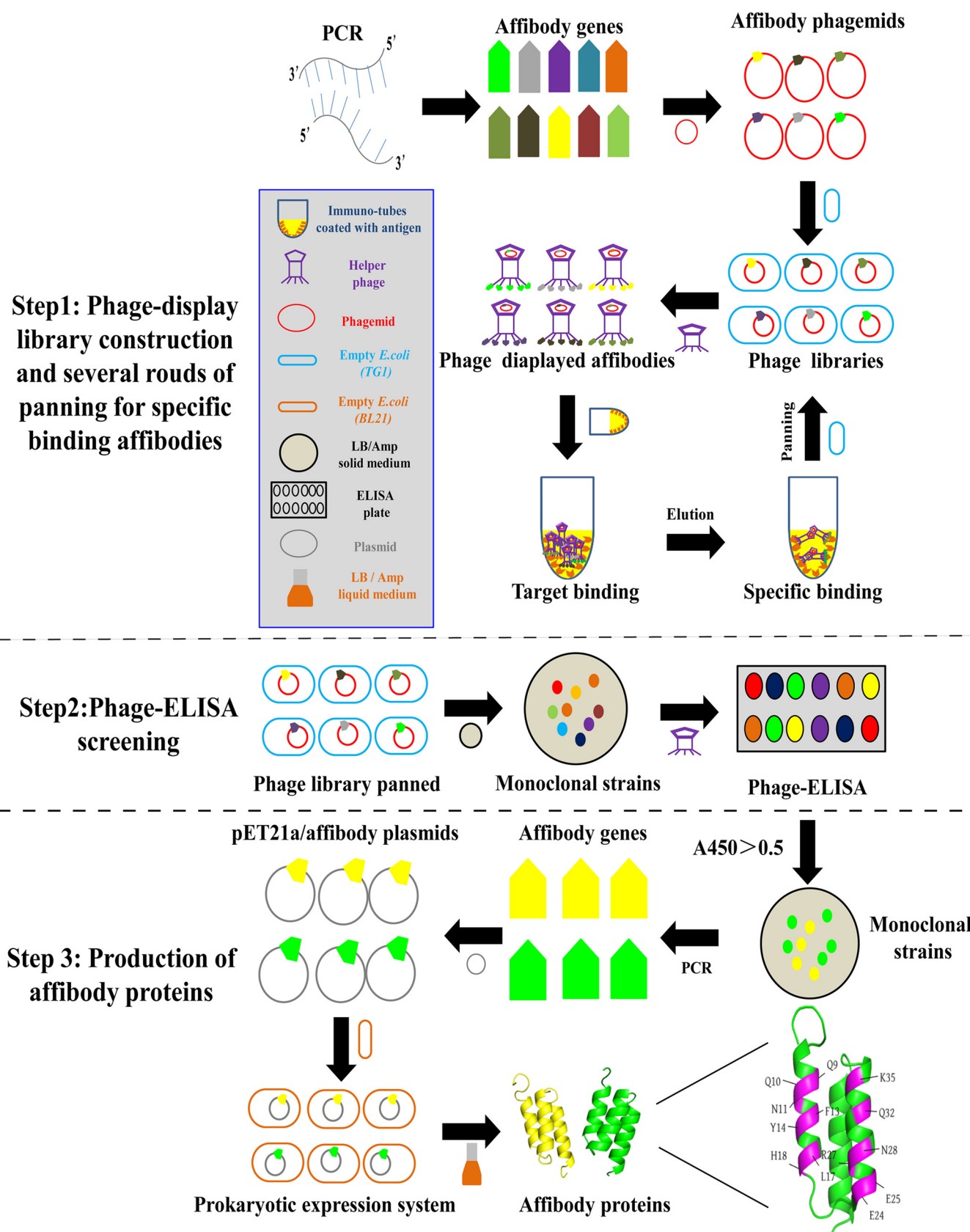

**FIG 1** Schematic procedures of specific binding affibody production. Step 1: phage-display library construction and several rounds of library panning (up to 20^13, affibody genes were obtained by PCR and constructed as affibody phagemids. Further subcloned into *E. coli* TG-1 bacteria, the primary library

affibody-targeted HER-2 has been approved in clinical usage for years and shows great potential for *in vivo* molecular imaging applications (42).

In this study, the recombinant SARS-CoV-2 RBMFP protein was expressed in *E .coli* first, and its immunogenicity and antigenicity were identified by mouse immunization and cell immunofluorescence assay. Subsequently, the RBMFP-binding affibody molecules were screened, and their affinity and neutralizing ability were characterized by surface plasmon resonance (SPR) and pseudovirus neutralization assay. Finally, the interaction sites of affibodies and the S protein were predicted by molecular docking. Our results indicated that the RBMFP protein is immunogenic and antigenic. We obtained four affibody molecules (Z14, Z149, Z171, and Z327) with high affinity to RBMFP and with neutralizing ability against SARS-CoV-2 prototype pseudovirus infection. Z14 and Z171 had neutralizing activities against SARS-CoV-2 Delta and Omicron pseudoviruses, which may be related to the interaction of F486 and Y489 residues of RBM. Here, we provide the first evidence that the affibody molecules are novel agents against pseudoviruses of SARS-CoV-2 variants.

## RESULTS

**Preparation of RBMFP fusion protein and characterization of RBMFP specific for ACE2 binding.** The schematic structures of the SARS-CoV-2 spike protein and recombinant RBMFP protein are shown in Fig. 2A. After molecular cloning (Fig. 2B) and identification by sequencing (see Fig. S1 in the supplemental material), the pET21a(+)/RBMFP plasmid was successfully constructed and transformed into *E. coli* BL21(DE3). The RBMFP expression was induced by isopropyl-$\beta$-D-thiogalactopyranoside (IPTG) and purified through nickel nitrilotriacetic acid (Ni-NTA) agarose resin. The purified recombinant RBMFP protein was resolved by SDS-PAGE (Fig. 2C) and identified by Western blot (Fig. 2D).

To evaluate whether the RBMFP protein has the ability to bind ACE2, we performed the immunofluorescence assay first to detect the RBMFP protein in colocalization with ACE2 in cells. The results (Fig. 3) showed that the ACE2 (anti-Flag, fluorescein isothiocyanate [FITC], green) was expressed in the HEK-293T-ACE2 cells (HEK-293T cells stably transfected with pLV-ACE2-3FLAG plasmids to overexpress human ACE2), whereas no ACE2 was detected in the HEK-293T cells (normally expressing no ACE2). ACE2 binding assays were conducted with the purified RBMFP expressed from *E. coli* by 100 $\mu$g/mL in final concentration for HEK293T-ACE2 cells for 3 h at 37°C along with HEK-293T cells as a control. Subsequent detection showed that the RBMFP protein (anti-His, Cy3, red) was colocalized (orange) with ACE2 in HEK-293T-ACE2 cells, but no such colocalization fluorescence signals were detected in the control HEK-293T cells. Data indicated that the RBMFP protein specifically binds ACE2.

**RBMFP protein elicited robust immunity in mice, and the serum neutralizes SARS-CoV-2 Omicron variant pseudovirus.** We assessed the immunogenicity of the recombinant RBMFP protein by mouse immunization and further detected high titers of the induced serum antibody specific for SARS-CoV-2 pseudovirus neutralization. The SARS-CoV-2 pseudoviruses were constructed by Tsingke Biotechnology (Hangzhou, China) using the lentiviral packing system, encoded the full-length sequence of SARS-CoV-2 spike proteins, and contained a *p*Lenti-green fluorescent protein (GFP) lentiviral reporter (15). The prepared pseudoviruses are replication deficient but infective for the ACE2-expressing cells using GFP and luciferase as reporters to be detected simultaneously.

After RBMFP immunization (Fig. 4A), the antibody titer of mouse serum increased (Fig. 4B), with a $1:10^5$ titer at the peak value on the 42nd day compared with the control group (Fig. 4C). After cells were infected with the pseudoviruses, GFP signals can be detected only in the HEK-293T-ACE2 cells, and the HEK-293T cells had no GFP signal detected (Fig. 4D), indicating that the SARS-CoV-2 pseudoviruses were able to

**FIG 1** Legend (Continued)
was constructed. After helper phages infected bacteria in libraries, affibodies were displayed on their surfaces, and through certain antigen binding and panning, the specific binding affibodies were panned into the following rounds of libraries). Step 2: phage-ELISA screening (monoclonal strains from the panned library were picked and screened using the same antigen). Step 3: production of affibody proteins (genes of specific binding affibodies were obtained through PCR and subcloned into a prokaryotic expression system for protein production).

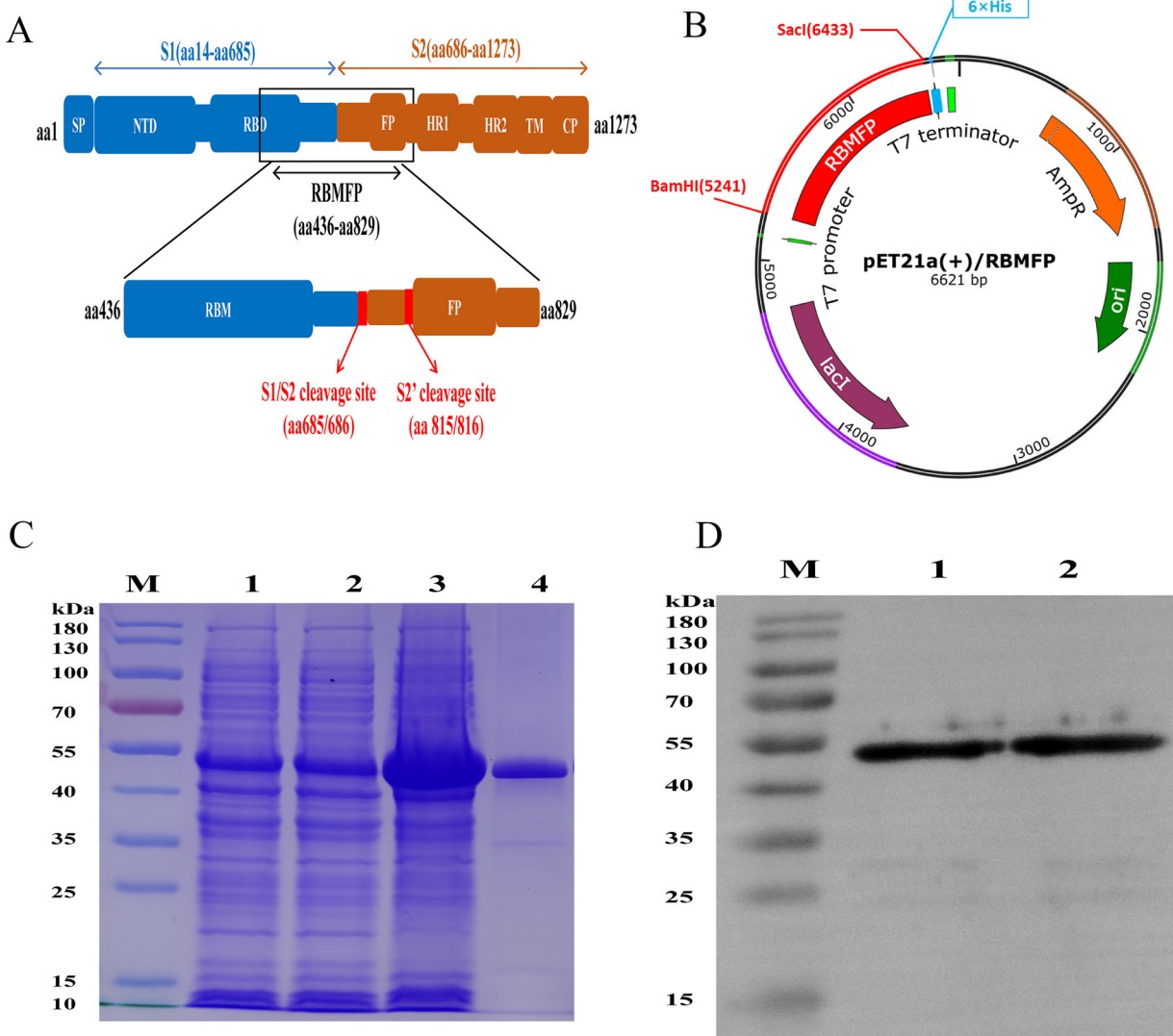

**FIG 2** Preparation and identification of the recombinant RBMFP protein. (A) Schematic structure of SARS-CoV-2 full-length spike protein and RBMFP recombinant protein. (B) Schematic structure of the pET21a(+)/RBMFP recombinant plasmid. (C) SDS-PAGE analysis identification of RBMFP. M, prestained protein ladder. Lanes 1 to 2, *E. coli* BL21(DE3) transformed with the recombinant pET21a(+)/RBMFP plasmid. Lane 3, *E. coli* BL21(DE3) transformed with the recombinant pET21a(+)/RBMFP plasmid induced by 1 mM IPTG. Lane 4, purified recombinant RBMFP protein. (D) Western blot analysis of RBMFP with anti-His tag as the primary antibody. Lane 1, *E. coli* BL21(DE3) transformed with pET21a (+)/RBMFP plasmid after being induced by 1 mM IPTG. Lane 2, Purified RBMFP protein. The concentrations of proteins used here were about 200 μg/mL.

infect cells with ACE2 expression. After an incubation with diluted serums, the infectivity of SARS-CoV-2 pseudoviruses decreased with increasing serum concentrations, as indicated by decreased GFP signals (Fig. 4E) and luminescent signals (see Fig. S2A in the supplemental material) detected in the HEK293T-ACE2 cells, and cell numbers were counted with no statistical significance (Fig. S2B), which showed that the sera induced by RBMFP could neutralize the pseudoviruses of Prototype and Omicron.

**Selection and preparation of RBMFP binding affibodies.** After 3 rounds of phage display panning, SARS-CoV-2 RBMFP-specific binding affibody clones (denoted $Z_{RBMFP}$ or ZN according to the clonal serial number "N" picked in this experiment) were selected and gathered in the tertiary library. Validated by using an LB/Amp(+) agar plate, the capacity of the tertiary library was confirmed up to $10^6$ (see Fig. S3 in the supplemental material). In total, 480 individual clonal strains were chosen from the library randomly and further selected by phage-enzyme-linked immunosorbent assay (ELISA) (see Fig. S4 in the supplemental material). The 100 clones with top A450 values were sequenced. Finally, 15 clones

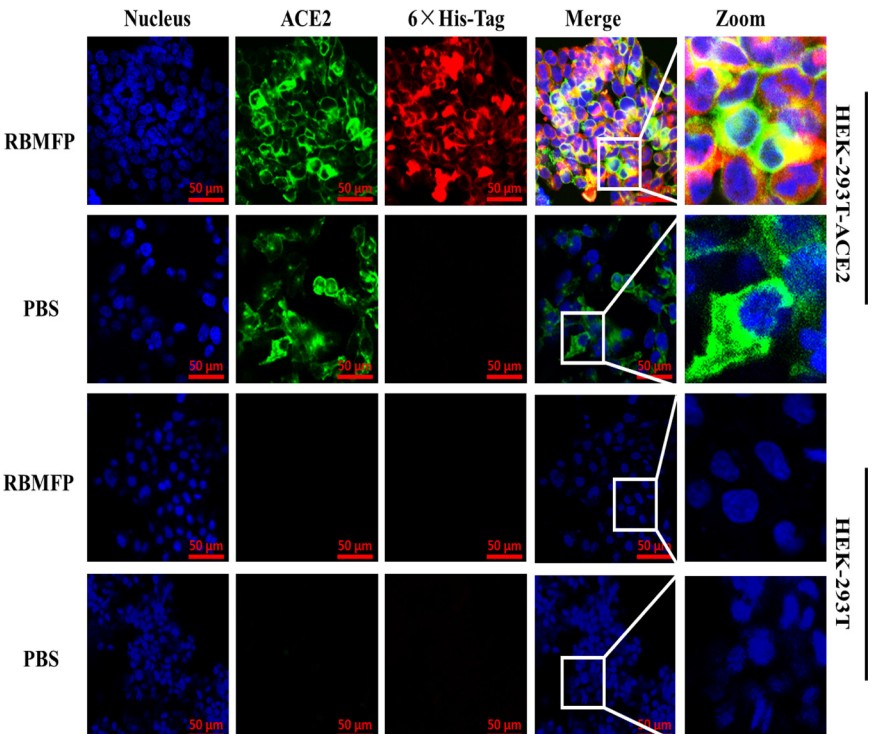

**FIG 3** Cellular immunofluorescence assay of RBMFP specifically binding to ACE2 expressed in cells. ACE2 (FITC, green) was expressed in HEK-293T-ACE2 cells only and the RBMFP protein (Cy3, red) specifically binding to ACE2 and colocalized (orange) with it in cells. The nuclei were stained by Hoechst 33258 (blue); scale bars, 50 $\mu$m.

that encoded the full-length sequences of affibody proteins were chosen, and their affibody genes were subcloned into the pET21a($+$) vector to construct the pET21a($+$)/$Z_{RBMFP}$ plasmids (Fig. 5A). After IPTG induction, the recombinant proteins of $Z_{RBMFP}$ and wild-type SPA-Z scaffold affibody (34) (Zwt) affibodies showed bands all at the molecular weight of about 7.5 kDa in SDS-PAGE (Fig. 5B). Subsequently, purified affibodies identified by SDS-PAGE analysis appeared in 7.5 kDa with high purity (Fig. 5C) and could be confirmed as the histidine-tagged affibodies with anti-His MAb (Fig. 5D).

**$Z_{RBMFP}$ affibodies bind SARS-CoV-2 RBMFP.** SPR was employed to analyze the binding affinity of $Z_{RBMFP}$ affibodies to the SARS-CoV-2 RBMFP protein. The results showed that the $Z_{RBMFP}$ affibodies interacted with SARS-CoV-2 RBMFP with different binding affinities (Fig. 6A). Z14, Z149, Z171, and Z327 were more prominent and were selected for further dynamic binding analysis. Their amino acid sequences were aligned along with Zwt (see Fig. S5 in the supplemental material). Resonance signals of gradient-diluted analytes from Z14 (Fig. 6C), Z149 (Fig. 6D), Z171 (Fig. 6E), and Z327 (Fig. 6F) showed concentration-dependent increases, and their dissociation equilibrium constants ($K_{D}$s) of them were about 94 to $\sim$660 nM (Table 1), which is significantly better than that of the Zwt affibody (Fig. 6B).

**$Z_{RBMFP}$ affibodies neutralize SARS-CoV-2 pseudovirus variants.** Compared with the Zwt SPA affibody, SARS-CoV-2 $Z_{RBMFP}$ affibodies Z14, Z149, Z171, and Z327 could effectively neutralize prototypic SARS-CoV-2 pseudovirus. The signals of both pseudovirus GFP (Fig. 7A) and luciferase (Fig. 7D) were decreased along with increasing $Z_{RBMFP}$ affibody doses, and the cell numbers were counted with no statical significance (see Fig. S6 in the supplemental material). Only Z14 and Z171 exhibited neutralization effects on the Delta variants (Fig. 8B and D) and even, surprisingly, neutralized the Omicron variants (Fig. 8C and D). In contrast, the Zwt affibody had no inhibition effects on the Delta or the Omicron variants. The half-maximal inhibitory concentration ($IC_{50}$)

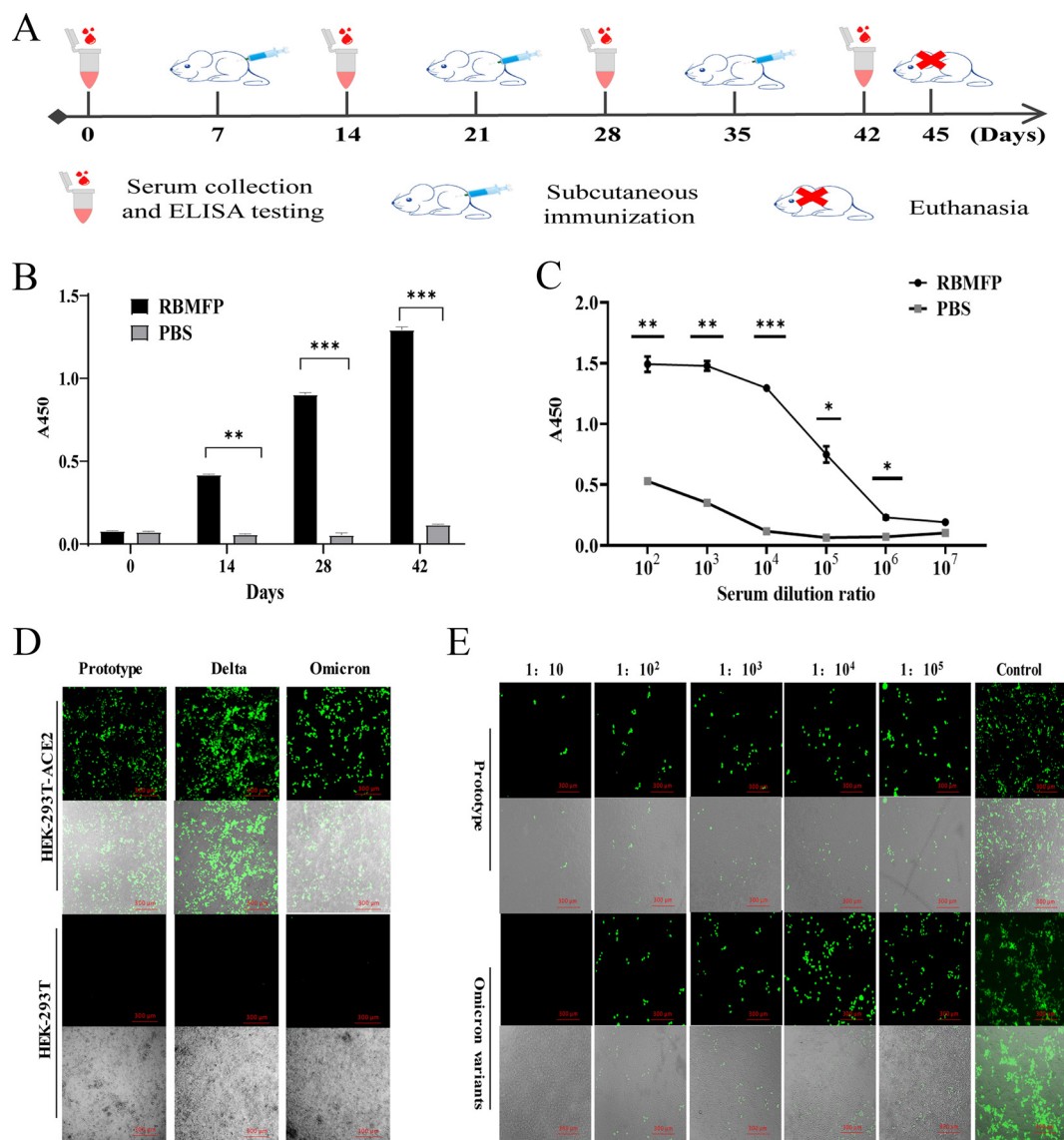

**FIG 4** Immunogenicity of recombinant RBMFP protein. (A) Timeline events of mouse immunization includes the following: days 7, 21, and 35 (immunization); days 0, 14, 28, and 42 (serum collection); and day 45, euthanasia. (B) Serum antibody (IgG) response of ELISA from immunized mice; the dilution ratio for detection is 1:10,000. RBMFP-induced antibody response enhanced over time, and there was a significant difference since the 14th day compared with the PBS immunization group (**, $P < 0.01$; ***, $P < 0.001$). (C) ELISA detection of serum antibody titer from the 42nd day after immunization (*, $P < 0.05$; **, $P < 0.01$; ***, $P < 0.001$). (D) SARS-CoV-2 pseudoviruses infected the HEK-293T-ACE2 cells and expressed GFP in cells. The HEK-293T cells without ACE2 expression had no GFP expression. (E) The neutralization effects of RBMFP-immunized mouse serum increased with concentration, indicating that mouse serum neutralizes SARS-CoV-2 pseudoviruses, including the Omicron variants.

values (Table 2) of $Z_{RBMFP}$ affibodies to neutralize the SARS-CoV-2 pseudovirus range from 1.12 to ~2.32 $\mu$M.

**Molecular docking.** Prediction models of molecular docking by cluspro service between affibodies and spike protein were about 80+ models per affibody, and we chose only the models that contain interaction sites between mutation sites of affibodies and the RBMFP region for analysis. Further, we reassessed cluster size and energy that were supplied by cluspro of those models selected, and the most promising models (see Fig. S7 in the supplemental material) with the largest cluster sizes of each affibody were chosen and showed by PyMol. The interaction sites of Z14 (Fig. 8A), Z149 (Fig. 8B), Z171 (Fig. 8C), and Z327 (Fig. 8D) with the S protein were shown, and their

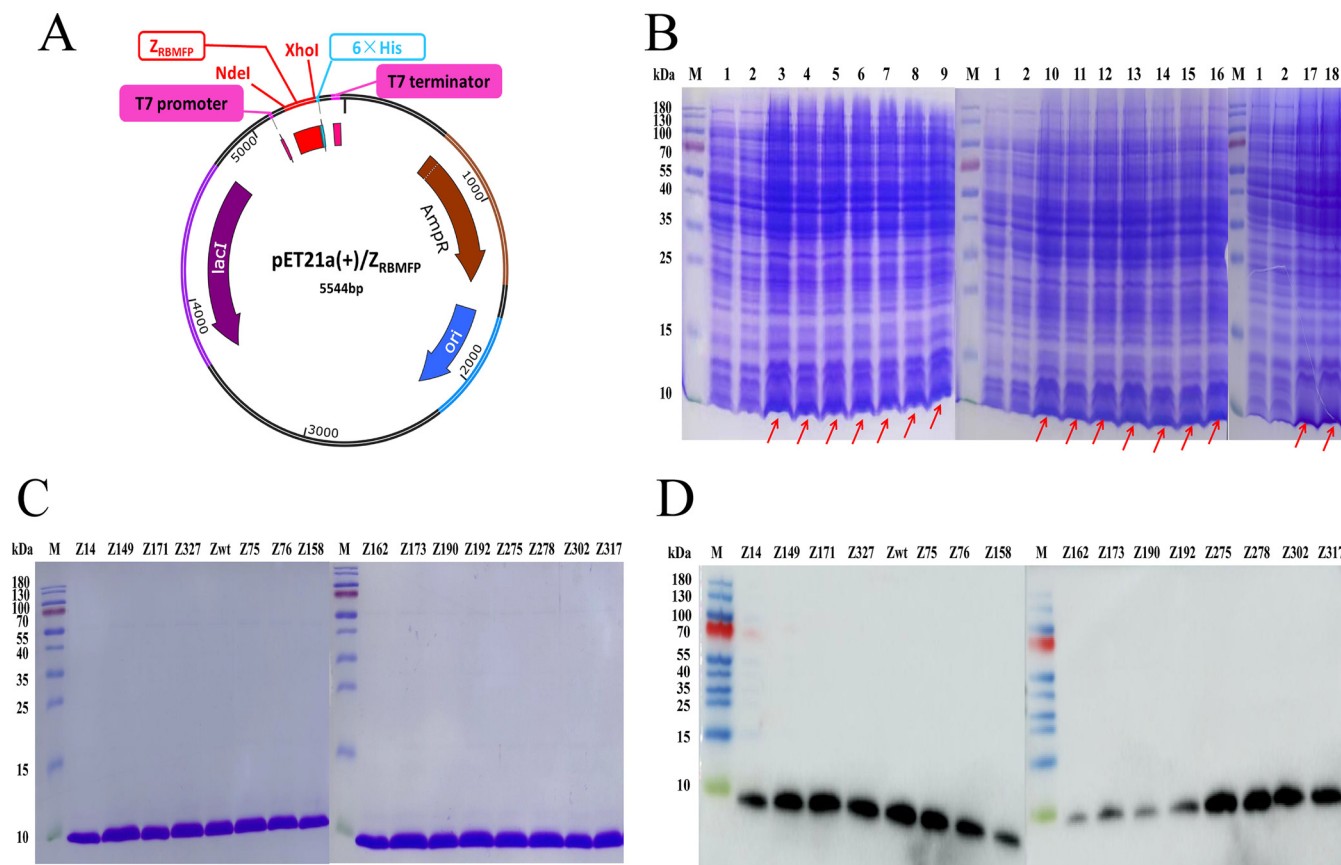

**FIG 5** Selection and preparation of RBMFP-binding affibodies. (A) Schematic structure of reconstructed pET21a(+)/Z$_{RBMFP}$ plasmids. (B) SDS-PAGE identification of the recombinant proteins expressed in *E. coli* BL21(DE3) under the induction with 1 mM IPTG. M, prestained protein ladder; lane 1, empty *E. coli* BL21(DE3) without plasmid; lane 2, *E. coli* BL21(DE3) transformed with pET21a(+) plasmid; lanes 3 to 9, 10 to 16, and 17 to 18, Z$_{RBMFP}$ expressed in *E. coli* BL21(DE3) transformed with pET21(+)/Z$_{RBMFP}$ plasmids (3, Z14; 4, Z149; 5, Z171; 6, Z327; 7, Zwt; 8, Z75; 9, Z76; 10, Z158; 11, Z162; 12, Z173; 13, Z190; 14, Z192; 15, Z275; 16, Z278; 17, Z302; 18, Z317). (C and D) SDS-PAGE (C) and Western blot (D) identification of purified Z$_{RBMFP}$. Lanes, purified Z$_{RBMFP}$. The concentration of proteins used for identification in each gel is 200 $\mu$g/mL.

alignments with ACE2 interaction sites were collated (Table 3). The results showed that the blockade of F486 and Y489 residues might be why Z14 and Z171 neutralize Delta and Omicron variants.

## DISCUSSION

The SARS-CoV-2 S protein is the major target antigen for developing vaccines and therapeutic MAbs. However, SARS-CoV-2 variants are emerging continuously and cause great challenges for the prevention, therapy, and control of SARS-CoV-2 infection (43). Here, we prepared and identified the RBMFP fusion protein, including RBM, FP domains, and cleavage sites from the S protein of the SARS-CoV-2 prototype strain, which is responsible mainly for virus binding (44) and entry (45) into the host cells. Our results showed that the RBMFP protein could specifically bind ACE2 expressed on the membrane of HEK-293T-ACE2 cells and elicited specific serum IgG antibodies in immunized mice. Moreover, the RBMFP-specific immunizing serum could neutralize both prototypic SARS-CoV-2 and its Omicron variant pseudovirus. The neutralization effects were dependent on serum concentrations, indicating that the antigenic protein from the prototypic S protein elicited a robust neutralizing antibody response against variants. Our data are consistent with what Qian et al. (46) reported. Therefore, the RBMFP protein could serve as an ideal target for our further affibody screening and even as a novel vaccine candidate.

The emergence of hybridoma technology and its application of MAbs brought a revolution in medicine. Since the 1970s, MAbs have been applied extensively in the

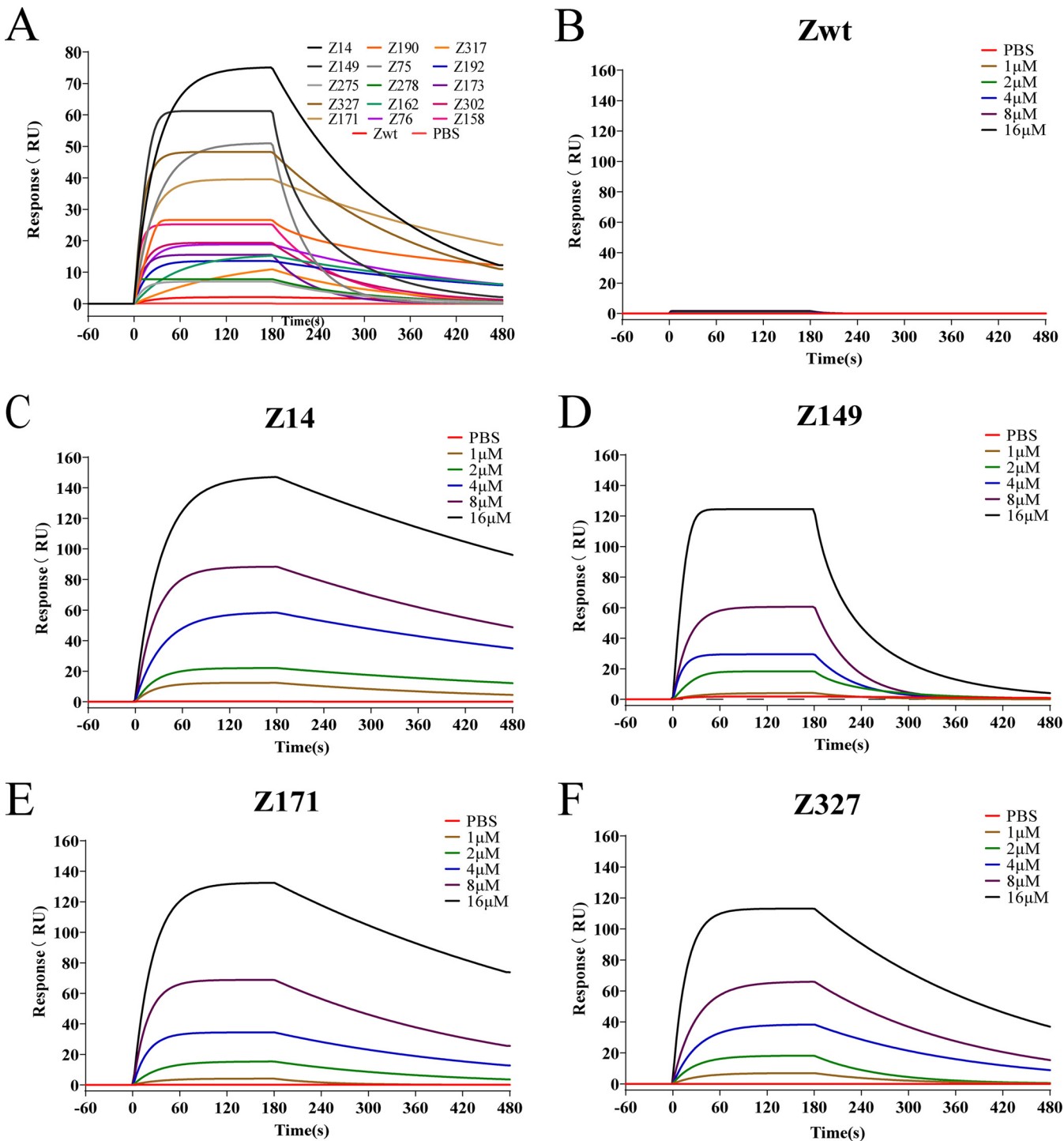

**FIG 6** SPR analysis of binding affinity between $Z_{RBMFP}$ and RBMFP protein. (A) Interaction resonance of $Z_{RBMFP}$ affibodies at the same concentration of 8 $\mu$M with purified recombinant RBMFP. (B to F) Binding responses of Z14, Z149, Z171, Z327, and Zwt at the gradient concentrations of 1, 2, 4, 8, and 16 $\mu$M interaction with the immobilized recombinant RBMFP protein.

fields of diagnosis, prevention, and treatment, and in studying immune mechanisms in human diseases. Moreover, the clinical use of MAbs targeted for CTLA-4, PD1/PD-L1 (47), *Staphylococcus aureus* alpha-toxin (48), and respiratory syncytial virus (RSV) F-protein (49) has created broad prospects for the immunodiagnosis and immunotherapy of human malignant tumors and infectious diseases. However, 2 years after SARS-CoV-2 was declared a global pandemic, a majority of novel MAbs against SARS-CoV-2 have

**TABLE 1** The kinetic binding constants of $Z_{RBMFP}$ interacting with the RBMFP protein

| Molecule | $K_a{}^a$ (1/Ms) | $K_d{}^b$ (1/s) | $K_D$ (M) |
|---|---|---|---|
| Z14 | $1.21 \pm 0.03 \times 10^4$ | $1.14 \pm 0.08 \times 10^{-3}$ | $9.41 \pm 0.69 \times 10^{-8}$ |
| Z149 | $7.66 \pm 1.67 \times 10^3$ | $2.38 \pm 0.44 \times 10^{-3}$ | $3.3 \pm 1.38 \times 10^{-7}$ |
| Z171 | $1.08 \pm 0.2 \times 10^4$ | $2.62 \pm 0.4 \times 10^{-3}$ | $2.44 \pm 0.08 \times 10^{-7}$ |
| Z327 | $3.64 \pm 0.22 \times 10^3$ | $2.38 \pm 0.37 \times 10^{-3}$ | $6.61 \pm 1.37 \times 10^{-7}$ |
| Zwt | $11.84 \pm 3.86$ | $0.363 \pm 0.22$ | $1.88 \pm 0.77 \times 10^{-2}$ |

[a]$K_a$, association constant.
[b]$K_d$, dissociation constant.

been discovered (50), but only a small portion of them are authorized for emergency use by the Food and Drug Administration (FDA), except for etesevimab and bamlanivimab, REGN10933/10987, AZD1061 and AZD8895, and S309 (5, 51). These MAbs were major isolated from B cells of patients and mainly targeted the S protein, with $IC_{50}$s for neutralization range from 4 pmol ~0.67 $\mu$mol (50), and individuals injected with three doses of vaccines remained to be susceptible to variants. The reason we guess is that the vaccines used today are based mainly on the original strain in which S proteins are major targets, and their protective effects against the variants declined with the S protein mutations (52). Moreover, since the N-glycan at N297 of IgG influences their binding to Fc$\gamma$ receptors (53), researchers designing therapeutic antibodies should consider the glycosylation issue and that they have challenges.

In contrast, an affibody is a small-molecule protein, only 6.5 kDa in size, with better penetration ability for tissue than an antibody (54). It could be prepared easily by prokaryotic expression without glycosylation and modified easily for molecular imaging and targeted therapy (38, 55). Our study identified that the prokaryotic expressed affibodies Z14, Z149, Z171, and Z327 could specifically bind the RBMFP protein with affinities of about 94 ~660 nM and neutralize the SARS-CoV-2 prototype pseudovirus. More importantly, Z14 and Z171 neutralized Delta and Omicron variant pseudovirus with $IC_{50}$s of 1.12 ~2.32 $\mu$M. However, the existing FDA-authorized MAbs that bind with RBD or for pseudovirus neutralization (such as bamlanivimab, $IC_{50}$ of 0.08 nM and $K_D$ of 3.5 nM; etesevimab [56], $IC_{50}$ of 0.24 nM and $K_D$ of 4.1 nM) have an advantage over the affibody. The reason for this finding may be that their larger size has more mutable sites for binding and better spatial steric resistance after binding. Fortunately, through constructing a dimer/trimer affibody or further mutating other certain residues in the affibody scaffold (57), the affinity of affibody can be further improved, and the larger size of the dimer or trimer will further improve their spatial steric resistance in the neutralizations. Otherwise, the cocktail therapy mixtures of the affibody with MAbs, or different affibodies together may be another promising strategy.

The mutation sites in the Delta variants possess 2 aa substitutions from the prototype RBD, which are L452R and T478K. Meanwhile, the Omicron variants possess 10 substitutions which are N440K, G446S, S477N, T478K, E484A, Q493R, G496S, Q498R, N501Y, and Y505H (58). According to the results of molecular docking and neutralization, we found that Z14, Z149, Z171, and Z327 all have neutralization effects on the prototype and bind directly to the G446, G476, E484, S494, F486, and Y505 residues in RBM, which are overlapped with the interaction sites of ACE2-RBM. Considering that Z14 and Z171 remain neutralizing Delta and Omicron variants, their binding residues are F486 and Y489 separately in the RBM of all three spikes, and the hydrophobic interactions contributed by F486/Y489 in the T470-F490 loop of spike protein are quite crucial for the ACE2 and RBM binding (59). Interestingly, the MAbs WRAIR-2125 (60), P5-22 (61), and XMA01 (62) are all effective to the Omicron variants and target residue F486 in RBM. Therefore, the residues F486 and Y489 may be the key residue sites for Omicron neutralization.

In conclusion, we developed four affibody molecules targeting SARS-CoV-2 S protein by phage display technology and provided the first evidence that they not only bind RBM with high affinity but also neutralize the prototype of SARS-CoV-2 pseudovirus. Especially, Z14 and Z171 could neutralize pseudoviral SARS-CoV-2 Delta and Omicron variants.

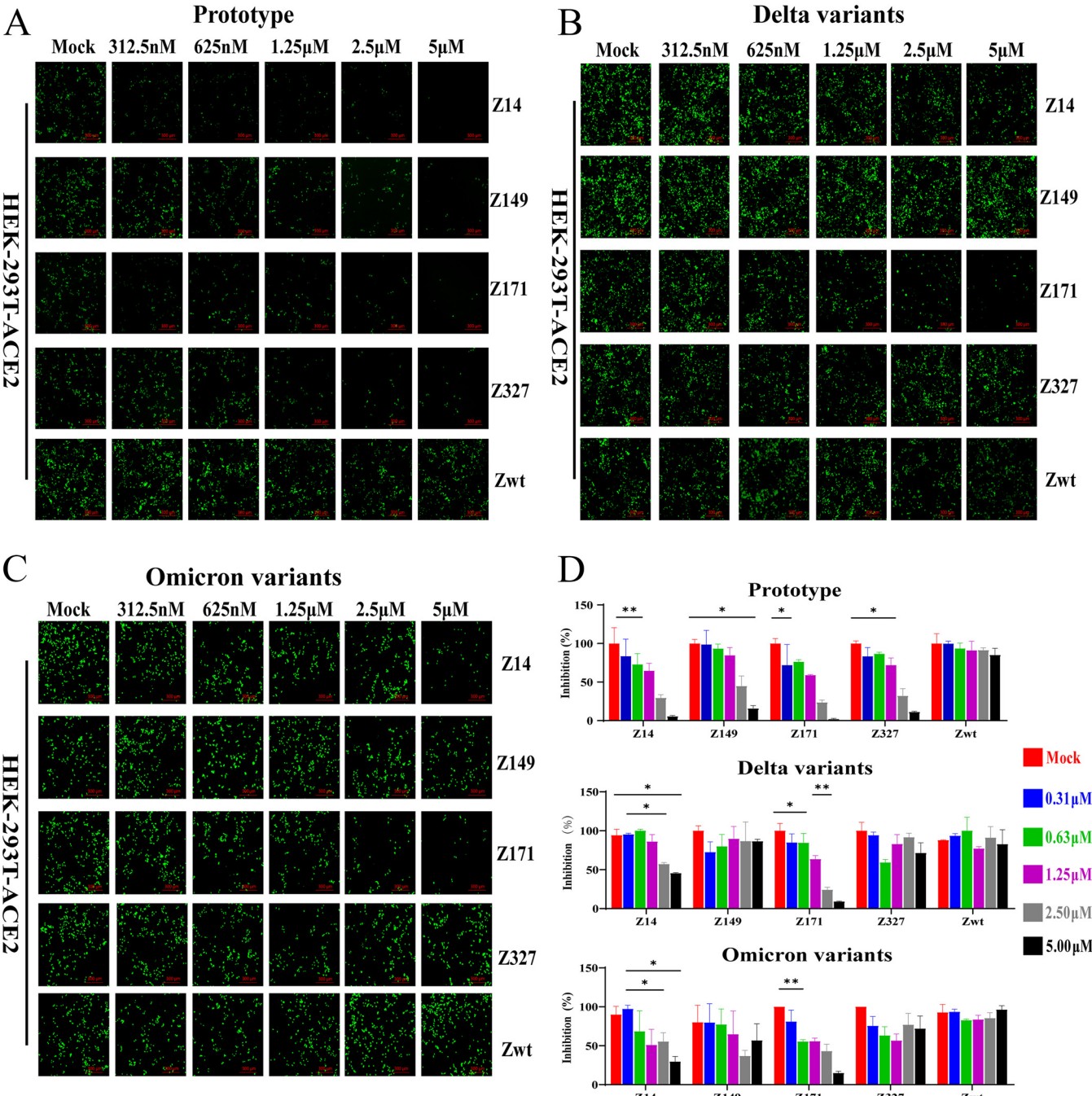

**FIG 7** Neutralization effects of $Z_{RBMFP}$ and Zwt to SARS-CoV-2 pseudoviruses. (A) GFP signals detected in the SARS CoV-2 pseudovirus prototype-infected HEK-293T-ACE2 cells in the presence of the indicated $Z_{RBMFP}$ affibodies or a Zwt affibody (scale bar, 300 $\mu$m). (B) GFP signals detected in SARS-CoV-2 pseudovirus Delta variant-infected HEK-293T-ACE2 cells in the presence of the indicated $Z_{RBMFP}$ affibodies or a Zwt affibody (scale bar, 300 $\mu$m). (C) GFP signals detected in SARS-CoV-2 pseudovirus Omicron variant-infected HEK-293T-ACE2 cells in the presence of the indicated $Z_{RBMFP}$ affibodies or a Zwt affibody (scale bar, 300 $\mu$m). (D) Luminescent light signals of luciferase were detected after neutralization and the inhibitory ratio calculated of $Z_{RBMFP}$ and Zwt to SARS-CoV-2 pseudoviruses (*, $P < 0.05$; **, $P < 0.01$).

However, more investigations on authentic virus neutralization, *in vivo* prevention, and crystal structure analysis should be carried out in future studies.

## MATERIALS AND METHODS

**Cell lines, plasmids, and reagents.** HEK-293T and HEK-293T-ACE2 cell lines were obtained from Tsingke Biotechnology Co., Ltd. (Hangzhou, China). The cells were all cultured in Dulbecco's modified Eagle's

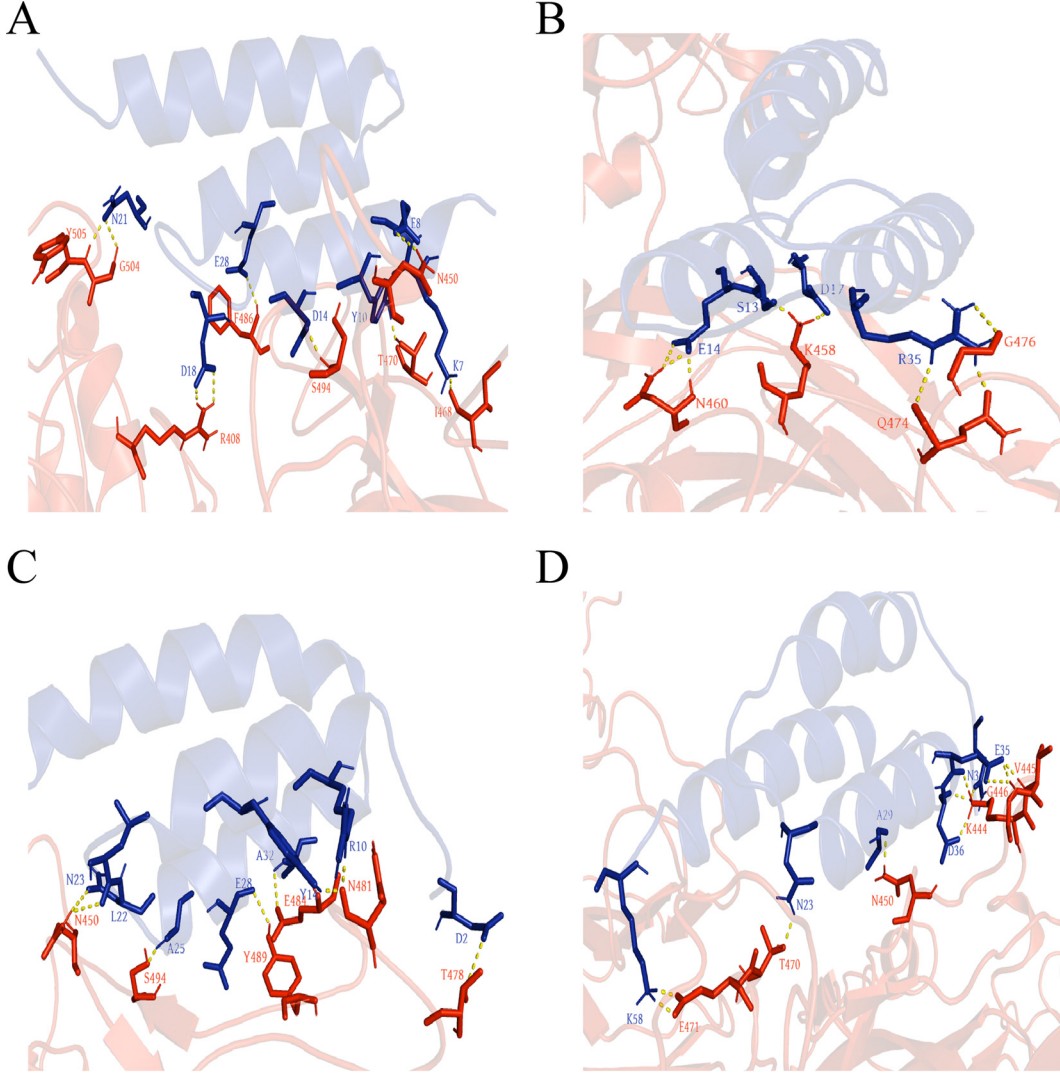

**FIG 8** Interaction sites between the RBM of SARS-CoV-2 S protein (PBD 7DZW) and $Z_{RBMFP}$. ($Z_{RBMFP}$ were shown in blue, the S protein were shown in red, and the polar contacts between $Z_{RBMFP}$ and S protein were shown in yellow).

medium (DMEM; Gibco, Invitrogen, USA) containing 10% fetal bovine serum (FBS; Gibco) and 1% penicillin-streptomycin (Gibco) at 37°C with 5% $CO_2$ supplied. pET21a(+) plasmid, pET21a(+)/Zwt plasmids, and *E. coli* BL21(DE3) strain were stocks in our laboratory. We also used *E. coli* TG1 strain, pCANTAB5E phagemid vector, and mouse anti-M13K MAb from the Bio-Viewshine company (Beijing, China) and restriction endonucleases (NdeI and XhoI) and T4 DNA ligase (New England BioLabs, MA). Isopropyl-D-thiogalactopyranoside (IPTG), paraformaldehyde, Triton X-100, ampicillin, kanamycin, and defatted milk powder were all products of Sigma (Saint Louis, USA). Ni-NTA agarose was from Qiagen (Dusseldorf, Germany). Bicinconinic acid (BCA) kit was from Beyotime (Beijing, China). PCR master mix was purchased from Tiangen company (Beijing, China).

**Preparation and characterization of recombinant RBMFP protein.** The gene sequence of the SARS-CoV-2 spike protein (GenBank MN908947.3) was obtained from the NCBI database (https://www.ncbi.nlm.nih.gov). After codon optimization for prokaryotic expression, the DNA of RBMFP was synthesized with 6×His-Tag added to the C terminus (Tsingke Biotechnology Co., Ltd. Hangzhou, China) and

**TABLE 2** IC$_{50}$ values of $Z_{RBMFP}$ and Zwt neutralize SARS-CoV-2 pseudovirus

| SARS-CoV-2 pseudovirus | IC$_{50}$ ($\mu$M) value of: | | | | |
| --- | --- | --- | --- | --- | --- |
| | Z14 | Z149 | Z171 | Z327 | Zwt |
| Prototype | 1.67 ± 0.067 | 2.0 ± 0.22 | 1.13 ± 0.03 | 1.4 ± 0.14 | |
| Delta variants | 1.84 ± 0.16 | | 1.32 ± 0.19 | | |
| Omicron variants | 2.32 ± 0.14 | | 1.12 ± 0.12 | | |

**TABLE 3** The interaction sites between RBM[a] and $Z_{RBMFP}$[b] overlapped or sterically interfered with the ACE2 receptor[c]

| Spike type | ACE2 | Binding sites overlapped | | | | Binding sites hindrance | | | |
|---|---|---|---|---|---|---|---|---|---|
| | | Z14 | Z149 | Z171 | Z327 | Z14 | Z149 | Z171 | Z327 |
| Prototype | K417, **G446**, Y449, Y453, L455, F456, A475, **G476**, **E484**, **F486**, N487, **Y489**, F490, Q493, G496, Q498, T500, N501, G502, **Y505** | F486, Y505 | G476 | E484, Y489 | G446 | Y449, E484, N487, Q493, G496, G502 | F456, A475 | Y449, G476, N487, F490, G496 | Y449 |
| Delta | K417, **G446**, Y449, Y453, L455, F456, A475, **G476**, **F486**, N487, **Y489**, Q493, G496, Q498, T500, N501, G502, **Y505** | F486, Y505 | G476 | Y489 | G446 | Y449, N487, Q493, G496, G502 | F456, Y449, A475 | Y449, G476, N487, Q493, G496 | Y449 |
| Omicron | Y449, Y453, L455, F456, A475, **G476**, N477, **F486**, N487, **Y489**, F490, R493, **S494**, Y495, S496, R498, T500, Y501, G502, H505 | F486, S494 | G476 | Y489, S494 | | Y449, N487, R493, Y495, G502, H505 | Y449, A475, N477 | Y449, N477, F486, N487, R493, Y495 | Y449 |

[a]PDB 7DZW.

[b]The interaction sites of $Z_{RBMFP}$ with SARS-CoV-2 spike protein (RBM) overlapped (bold face) and sterically hindered (underlined) the ACE2 binding. The mutation sites of RBM in Delta variants are L452R and T478K; in Omicron, variants are N440K, G446S, S477N, T478K, E484A, Q493R, G496S, Q498R, N501Y, and Y505H (41).

[c]Binding sites of ACE2 receptor with RBM of SARS-CoV-2 Prototype, Delta, and Omicron.

inserted into the pET21a(+) expression vector through SacI and BamHI restriction sites to construct the pET21a(+)/RBMFP recombinant plasmid. Further identified by sequencing, the reconstructed pET21a (+)/RBMFP plasmids were transformed into the *E. coli* BL21(DE3) system and induced by 1 mM IPTG at 37°C for 6 h to express the fusion protein. Then the bacterial sediments were harvested and dissolved in 8 M urea for ultrasonic lysis, and the supernatants were collected after centrifugation. The recombinant RBMFP protein with the 6×His tag was purified through Ni-NTA affinity chromatography and washed out with an elution buffer. Through gradient dialysis refolding, the purified RBMFP protein dissolved in phosphate-buffered saline (PBS) (Solarbio, Beijing, China) was verified by SDS-PAGE and further confirmed by Western blot using the mouse anti-His MAb (MultiSciences, Hangzhou, China). The protein concentration was tested by the bicinchoninic acid (BCA) method, aliquoted, and stored at −80°C for further use.

**Colocalization of immunofluorescence.** To confirm whether the recombinant RBMFP protein had the ability to recognize human ACE2, the indirect immunofluorescence assay method was developed. First, HEK-293T-ACE2 or HEK-293T cells were cultured on sterile slides in a 6-well cell culture plate at 37°C for 24 h. After being incubated with RBMFP protein at a final concentration of 100 $\mu$g/mL or with PBS at the same volume for 3 h in a cell incubator, slides were washed twice with PBS and fixed with 4% paraformaldehyde at 37°C for 10 min. Subsequently, Triton X-100 (0.3%) was used to permeabilize the cells for 10 min, and the cells were blocked by DMEM containing 10% FBS. Then, cells were incubated simultaneously with 2 different primary antibodies (mouse anti-Flag MAb, 1:2,000, MultiSciences; rabbit anti-His MAb, 1:2,000, Beyotime) at 4°C overnight and then washed with PBS with Tween 20 (PBST). The secondary antibodies (Cy3-conjugated goat anti-mouse MAb, 1:2,000, Beyotime; FITC-conjugated goat anti-rabbit MAb, 1:2,000, MultiSciences) were added simultaneously to bind the primary antibodies at 37°C for 2 h and then washed with PBST. The nuclei of cells were stained with Hoechst 33258 at 37°C for 10 min before fluorescence was detected by laser scanning confocal microscopy (FV3000; Olympus).

**Mouse immunization and serum collection.** To identify the immunogenicity of RBMFP, we immunized the mice and tested their antibody titer. The experiment protocols were described previously with minor changes (63). Briefly, 10 female BALB/c mice of 6 to 8 weeks old were divided randomly into 2 average groups, as follows: one group was immunized subcutaneously with 50 $\mu$g RBMFP protein in 100 $\mu$L sterile PBS and the other with 100 $\mu$L sterile PBS only. Proteins were mixed with 100 $\mu$L complete

Freund's adjuvant in the primary immunization (days 7), and the following immunizations (days 21 and 35) were both with 100 $\mu$L incomplete Freund's adjuvant. Serums of days 0, 14, 28, and 42 were collected separately.

**Mice serological ELISA.** The antibody titer of serums was identified by ELISA with RBMFP being coated as an antigen and horseradish peroxidase (HRP)-conjugated goat anti-mouse IgG MAb being used as a secondary antibody. First, purified RBMFP protein in carbonate coating buffer at a final concentration of 10 $\mu$g/mL was coated at 100 $\mu$L/well on the 96-well ELISA plates at 4°C overnight, and the unbound molecules were then washed 3 times by PBST. After a blocking step with 5% defatted milk at 37°C for 2 h and 3 PBST rinses were performed, serums from each week that were diluted in PBS with a 10 times gradient ratio ($10^2$ $\sim$$10^7$) were added at 100 $\mu$L/well and incubated at 37°C for 2 h. Then the plates were washed 3 times with PBST. After the wash step, the HRP-conjugated goat anti-mouse IgG MAb (1:5,000; MultiScience, Hangzhou, China) was added at 100 $\mu$L/well, reacted at 37°C for 2 h, and then rinsed with PBST. Then, the chromogenic tetramethylbenzidine (TMB; Innoreagents, Huzhou, China) solution was added at 100 $\mu$L/well and incubated at 37°C for 10 min. Finally, 50 $\mu$L/well of the ELISA stop solution (Beyotime, Beijing, China) was added, and the absorbance (optical density [OD]) at 450 nm was measured in a Bio-Tek ELISA microplate reader. The antibody titer of the serums was evaluated and statistically analyzed by GraphPad Prism 8.0 software.

**Serum neutralization for pseudovirus.** Serum from immunized mice was filtered through a 0.22-$\mu$m filter and diluted into 5 gradient concentrations (1:10, 1:$10^2$, 1:$10^3$, 1:$10^4$, and 1:$10^5$) with the complete culture medium, and the complete culture medium with no mice serum was set as a negative control. First, 98 $\mu$L negative-control medium was premixed with 2 $\mu$L SARS-CoV-2 pseudoviruses (titers: 2 $\times$ $10^7$ TU/mL) separately and incubated at 37°C for 1 h. A total of 1.0 $\times$ $10^4$ cells/well of HEK-293T-ACE2 and HEK-293T cells were seeded separately in a 96-well plate and cultured for 24 h before infection, and the medium in the plate was removed and washed with PBS. Then, the pseudovirus mixtures were added to 100 $\mu$L/well and cultured for 8 h with cells. After that step, the medium was removed, cells were rinsed with PBS, 100 $\mu$L fresh complete culture medium was added to each well, and cells were cultured for another 48 h. The GFP signals expressed in cells were detected by an inverted fluorescence microscope (DMi8; Leica), and photos were taken. The luciferase signals were detected with a Varioskan LUX instrument (ThermoFisher, Waltham, USA) by detecting the luminescent light after cells were handled with the assay kit (Yeasen, Shanghai, China) according to the instructions in the manual. To identify the mice serum neutralization for pseudovirus, 98 $\mu$L diluted mice serums and negative control were premixed with 2 $\mu$L SARS-CoV-2 pseudoviruses separately and incubated at 37°C for 1 h, and the following procedures were the same as described above but were performed only on HEK-293T-ACE2 cells. The neutralization ability of the serum was evaluated by estimating the number of pseudovirus-infected cells after being premixed. The better the neutralization results, the fewer GFP or luciferase signals would be detected in cells. Cell numbers in each picture were counted in Image Pro Plus 6.0 software, and statistical analysis was conducted using GraphPad Prism 8.0.

**Phage display panning for RBMFP targeting affibodies.** A phage display library with a combinatorial of 1 $\times$ $10^{12}$ affibody clones was constructed first (34), and the panning of RBMFP binding affibodies was performed as described previously (37). Briefly, 3 rounds of phage display panning were conducted with the purified RBMFP protein coated in immune tubes as the antigen, and then the RBMFP-specific affibodies displayed on the surface of helper phages were obtained in *E. coli* TG-1 bacteria and clustered in the tertiary RBMFP-binding phage library. Picking up single colonies from the tertiary phage library on an LB/Amp(+) solid medium plate numbered each colony separately. Finally, 480 clones were cultured with phages, and their supernatants were taken for ELISA screening.

**Phage-based ELISA screening.** In ELISA screening, the purified RBMFP protein in carbonate coating buffer was coated on the 96-well ELISA plates at 4°C overnight, and the unbound molecules were then washed 3 times by PBST. After being blocked with 5% defatted milk at 37°C for 2 h and rinsed with PBST 3 times, the 480 supernatants of cultures were added separately at 100 $\mu$L/well for binding with RBMFP at 37°C for 2 h. Then the plates were washed 3 times with PBST before the mouse anti-M13K MAb (1:5,000; Bio-Viewshine, Beijing, China) was added at 100 $\mu$L/well to incubate at 37°C for 2 h. After the wash step, the HRP-conjugated goat anti-mouse IgG MAb (1:5,000; MultiScience, Hangzhou, China) was added at 100 $\mu$L/well to react at 37°C for 1 h and then rinsed with PBST. The chromogenic tetramethylbenzidine (TMB; Innoreagents, Huzhou, China) solution was added to 100 $\mu$L/well and incubated at 37°C for 10 min. Finally, each well was added with 50 $\mu$L of the ELISA stop solution (Beyotime, Beijing, China), and the OD at 450 nm ($OD_{450}$) was measured in a Bio-Tek ELISA microplate reader. The corresponding clones with higher signals of the $OD_{450}$ absorbance value were selected and further used after sequencing correctly.

**Production of affibody molecules.** The DNA of 15 screened $Z_{RBMFP}$s was amplified by PCR (34), and a 6×His-tag was fused to the C terminus simultaneously. Through NdeI and XhoI restriction sites, the fragments were subcloned into the pET21a(+) vector separately to construct the recombinant plasmids pET21a (+)/$Z_{RBMFP}$. After sequencing, pET21a(+)/$Z_{RBMFP}$ and pET21a (+)/Zwt were transformed into *E. coli* BL21(DE3) and induced by 1 mM IPTG for 6 h at 37°C to express the affibody fusion proteins. After ultrasonic disruption of bacteria and purification by Ni-NTA Sepharose column with different concentrations of imidazole gradient diluted, purified $Z_{RBMFP}$ proteins (Z14, Z149, Z171, Z327, Zwt, Z75, Z76, Z158, Z162, Z173, Z190, Z192, Z275, Z278, Z302, and Z317) were collected. After being verified by SDS-PAGE and Western blot analysis using mouse anti-His MAb as the primary antibody, the concentrations of proteins were detected using the BCA method, aliquoted, and stored at −80°C for further use.

**Surface plasmon resonance (SPR).** SPR was performed on a BIAcore X100 (GE Healthcare, Uppsala, Sweden) instrument. The CM5 sensor chip was immobilized with the RBMFP protein using 1-ethyl-3-(3-dimethylaminopropyl) carbodiimide hydrochloride/N-hydroxysuccinimide (EDC/NHS) coupling reagent first, and then the purified proteins $Z_{RBMFP}$ and Zwt were injected separately over the flow-cell surfaces

of the CM5 sensor chip at the same concentration. After primary selection, Z14, Z149, Z171, and Z327 with the maximum response unit (RU) were selected and further analyzed in various concentrations ranging from 1 ~16 $\mu$M with Zwt as the negative control. The results were all analyzed using a 1:1 binding model, and the data were fitted and evaluated via BiacoreX100 evaluation software on the instrument.

**Neutralization of Z$_{RBMFP}$ to SARS-CoV-2 pseudovirus.** The neutralization effect evaluation experiments of Z14, Z149, Z171, and Z327 to SARS-CoV-2 pseudovirus (Prototype, Delta, Omicron) were performed like the mouse immune serum neutralization described above. Briefly, $1.0 \times 10^4$ cells/well HEK-293T-ACE2 cells were first seeded and cultured for 24 h in a 96-well plate. Then Z$_{RBMFP}$ proteins in 5 gradient concentrations (0.3125 ~5 $\mu$M) together with the Zwt and PBS as negative control were premixed with SARS-CoV-2 pseudovirus (Prototype, Delta, Omicron) and incubated at 37°C for 1 h. After the mixtures were incubated with cells for 8 h, culture supernatants were removed, and after rinsing with PBS, the fresh complete culture medium was added, and the cells were further cultured for 48 h. After that, the fluorescent images of GFP in cells were taken, and the luciferase signals were detected. Cell numbers in each picture were counted in Image Pro Plus 6.0 software, and statical analysis was conducted using the GraphPad Prism 8.0 software. Finally, the half-maximal inhibitory concentrations (IC$_{50}$s) of Z$_{RBMFP}$ against the pseudoviruses were assessed using the GraphPad Prism 8.0 software with normalization with the negative control at the 100% level, and the inhibition rate of each sample were calculated.

**Molecular docking.** Molecular docking was performed on the cluspro2.0 service, which has been used widely for protein-protein docking (64). First, we simulated the spatial structures of Z14, Z149, Z171, and Z327 using the SWISS-MODEL (https://swissmodel.expasy.org/) with the other crystal structure of affibodies (SMTL IDs 5djt.1, lp1.1.A, 2m5a.1, and 2kzi.1A) as the templates. The receptor was from the RC-SB database (PDB 7dzw) and is the crystal structure of the SARS-CoV-2 spike protein with a D614G mutation derived from the prototype. After all other ligands, water molecules, and ions were removed, the prepared models were uploaded on the cluspro Web service (https://cluspro.bu.edu/) for molecular docking (64). The results were then downloaded and visualized by PyMol (Schrödinger) software. During the analysis, we selected the RBMFP domain in the receptor and the mutated residues (residues sites 9 to 11, 13 to 14, 17 to 18, 24 to 25, 27 to 28, 32, and 35) in Z$_{RBMFP}$ as interacted regions, and we analyzed the polar interactions between them. The models with polar interactions were chosen and further analyzed retrospectively with raw cluster and energy scores from cluspro. Finally, the models of each team with the largest cluster size based on primary screening were selected as the best model according to the instructions cluspro provided. The interaction sites of these models were further analyzed and shown in PyMol.

**Ethics statements.** Female BALB/c mice (6 to 8 weeks) were obtained from the Shanghai Experimental Animal Center of the Chinese Academy of Sciences (Shanghai, China), which were monitored and cared for strictly by following laboratory animal ethics and maintained in the Laboratory Animal Center of Wenzhou Medical University. The ethics are approved by the experimental animal ethics committee of Wenzhou Medical University, and the approved number is wydw2021-0585.

**Statistical analysis.** All experiments were performed three times. Data presented as the mean ± standard deviation (SD) were plotted by GraphPad Prism 8.0 software and statistically analyzed using analysis of variance (ANOVA), and a $P$ value of ≤0.05 was considered statistically significant.

## SUPPLEMENTAL MATERIAL

Supplemental material is available online only.
**SUPPLEMENTAL FILE 1**, PDF file, 0.08 MB.

## ACKNOWLEDGMENTS

We sincerely thank Zhiming Zheng, Tumor Virus RNA Biology Section, RNA Biology Laboratory, Center for Cancer Research, National Cancer Institute, National Institutes of Health, Frederick, Maryland, for his guidance in the study and proofreading of the manuscript.

W.D. conducted most of the experiments, performed data analysis, and mainly wrote the manuscript; P.J. assisted in the selection of novel affibodies, prepared proteins, and helped with the pseudovirus neutralization experiments; Q.L. constructed the phage library and immunized mice with W.F.; H.W., J.Z., M.Z., J.Y., and Y.G. assisted in the preparation of proteins; and S.Y. provided the fluorescence microscope and helped take photos; L.Z., W.L., S.Z., and S.K., wrote and revised the original draft together. L.Z., W.L., S.Z., P.J., and J.C. analyzed and guided the experiments. L.Z. and S.Z. provided the funds for all the experiments.

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
