## [Reviewer comments · Microbiology Spectrum]

Microbiology Spectrum

Novel Affibody Molecules Specifically Bind to SARS-CoV-2 Spike Protein and Efficiently Neutralize Delta and Omicron Variants

Du Wangqi, Jiang Peipei, Li qingfeng, Wen He, Zheng Maolin, Zhang Jing, Guo Yanru, Yang Jia, Feng Weixu, Ye Sisi, Saidu Kamara, Jiang Pengfei, Chen Jun, Li Wenshu, Zhu Shanli, and Lifang Zhang

Corresponding Author(s): Lifang Zhang, Wenzhou Medical University

Review Timeline:

Submission Date:	September 4, 2022
Editorial Decision:	October 14, 2022
Revision Received:	November 11, 2022
Accepted:	November 21, 2022

Editor: Yongjun Sui

Reviewer(s): Disclosure of reviewer identity is with reference to reviewer comments included in decision letter(s). The following individuals involved in review of your submission have agreed to reveal their identity: Zheng Chunfu (Reviewer #1)

Transaction Report:

DOI: <https://doi.org/10.1128/spectrum.03562-22>

October 14, 2022

Prof. Lifang Zhang
Institute of Molecular Virology and Immunology, Department of Medical Microbiology and Immunology
306 hualongqiao road
Wenzhou
China

Re: Spectrum03562-22 (Novel Affibody Molecules Specifically Bind to SARS-CoV-2 Spike Protein and Efficiently Neutralize Delta and Omicron Variants)

Dear Prof. Lifang Zhang:

Link Not Available

Sincerely,

Yongjun Sui

Journals Department
Reviewer comments:

Reviewer #1 (Comments for the Author):

The authors selected the domains of RBM and FP from SARS-CoV-2 S protein that binds and fuses with cell membrane receptors as targets, prepared the RBMFP fusion protein using a prokaryotic expression system, and identified its immunogenicity and antigenicity. Further, they selected the novel affibodies with phage-display technology and obtained affibodies with binding affinity to RBMFP. Then, the neutralization effect of affibodies on SARS-CoV-2 pseudovirus was evaluated.

The authors present compelling data to support the binding specificity of these affibodies and their neutralizing efficacy against SARS-CoV-2 pseudovirus. The paper is well written and interesting, especially Z14 and Z171 neutralize against Omicron

variants, a dominant epidemic today, so the researchers have potential and application prospects; however, some concerns have been raised during the revision process.

1. Please explain or evaluate affibodies compare with neutralizing antibodies against the SARS-CoV-2 virus relative to advantage. Is there any affibody being developed specifically for clinical utilization?
2. The authors used the RBMFP sequence origin of the SARS-CoV-2 prototype. Why not use the currently popular omicron? Are there any reasons for this?
3. The data in the manuscript demonstrates that the prokaryotic recombinant RBMFP protein is immunogenic, and its specific immune mouse serum also has the effect of neutralizing the pseudovirus of the new coronavirus variant. Compared with the currently marketed vaccines, how are they different?
4. The authors point out that they have produced "high-affinity" binders. The authors report single-digit micromolar KDs calculated based on the binding kinetics using SPR. Although there is such clear cell binding and measurable in vitro, a binder with μM affinity may not be ideal for the preventive or therapeutic agent. Whether further improvements are needed, or how to modify them further to increase their affinity for further development and application?
5. Molecular docking predicts the preferred orientation, affinity, and interaction of a ligand in the binding site of a protein. Could the authors explain the binding sites of Z14 and Z171 affibody molecules?
6. Common English language errors, for example, Fig. 2 and 3 legends. Overall, the text is good. Just some minor corrections are needed in the manuscript.

Reviewer #2 (Comments for the Author):

Overview: This manuscript provides an interesting development narrative of four novel affibodies targeting the RBMFP of the SARS-CoV-2 spike protein. Effort was taken to evaluate the antigenicity of the RBMFP protein and understand whether affibodies produced by a phage display platform are capable of neutralizing SARS-CoV-2 variants. Although there is a data to support some of the conclusions made in this manuscript, this reviewer would like to address several challenges:

Major comments:

The neutralization data in this study are not well supported. Single fluorescent images are provided and neutralization extrapolated without an understanding of cell density in each image or appropriate controls in some cases (see detailed comments). Without this information it is extremely difficult to validate the presence of true neutralization and the major emphasis of this manuscript.

This reviewer would appreciate additional evidence of affibody binding to endogenous respiratory cell lines that express human ACE2 to ensure that binding remains in an unmodified / biologically relevant cell line.

Detailed comments:

Single references of critical statements are inadequate to support many statements that can be validated by multiple studies at this point in the pandemic.

There is very little methodological description of the serological ELISA that was performed and would not be reproducible from this method. This should be described in much better detail.

Figure 4C: Statistical analysis of titer, area under the curve, EC50 etc. would help provide additional support for this conclusion.

Figure 4D-E: Serum neutralization should be validated with quantification of fluorescent intensity or by additional surrogate assays. Single images demonstrate a potential trend however this does not adequately support the conclusion that all RBMFP+ sera provides a neutralizing effect. In addition, the PBS group is not represented in either Figure 4D and 4E which does not provide a point of comparison and no phase contrast images providing a reference for cell number per image is provided.

Figure 5C-D: Labeling of the gels and blots would help provide additional clarity to the figure.

Figure 7: Phase contrast images should be provided and luminescent / fluorescent intensity values should be normalized to the area of cells per well. Without normalizing to cell number / area per image it is possible that a reduction in luminescent / fluorescent intensity is merely an artifact of fewer cells per image. Additionally, a statement on the number of images analyzed per sample should be provided.

Figure 7D: Statistical analysis of multiple groups should be performed using ANOVA, not t-test.

Lines 56-57: There is no evidence provided to suggest that the RBMFP provides neutralizing protection in vivo in this manuscript

Line 69: Reference number seven only refers to vaccine resistance of the delta variant. This statement should be qualified by additional studies demonstrating viral resistance to existing vaccines.

Lines 61-71: Statements in introduction should be referenced in more detail. Many studies exist supporting this logic and single references do not adequately describe the literature encompassing this topic.

Lines 73-79: References should be included to support these statements.

Lines 99-101: Further references regarding the structure, function, and utility of affibodies would be beneficial. For example: Frejd, F., Kim, KT. Affibody molecules as engineered protein drugs. *Exp Mol Med* 49, e306 (2017).

<https://doi.org/10.1038/emm.2017.35>

Lines 127-128: Is this statement supported by data in the manuscript? Neutralizing capacity in vitro does not necessarily

translate to in vivo.

Line 178: What does "correct sequence" mean?

Line 225: Reference the mechanism of SARS-CoV-2 entry to cells.

Line 227: "High level of specific...". This phrasing is not valid. No comparison for whether this degree of IgG production qualifies as high. Please revise by comparing to a control or remove "high level".

Lines 253-254: This reviewer would appreciate clarity from the author's as to whether there is evidence an advantage of affibodies against heavily glycosylated proteins over mAbs. If there is no evidence to support this, this statement should be revised to reflect this.

Lines 247, 257-262: The KD of bamlanivimab is 0.071nM and blocks ACE2 spike interactions with an IC50 of 0.17nM. The KD of estevimab is 6.45nM and blocks ACE2-spike interactions with an IC50 of 0.32nM. It would be useful to put the affinity and neutralizing capacity of the affibodies in direct context with these existing FDA-authorized drugs for discussion. The values provided in line 247 are misleading if not in the same units as the affibody molecules. KD and IC50 values of affibody molecules greater than 10-fold higher than existing drugs should be discussed and statements regarding prospective efficacy tempered accordingly.

Staff Comments:

Preparing Revision Guidelines

Please return the manuscript within 60 days; if you cannot complete the modification within this time period, please contact me. If you do not wish to modify the manuscript and prefer to submit it to another journal, please notify me of your decision immediately so that the manuscript may be formally withdrawn from consideration by Microbiology Spectrum.

Overview: This manuscript provides an interesting development narrative of four novel affibodies targeting the RBMFP of the SARS-CoV-2 spike protein. Effort was taken to evaluate the antigenicity of the RBMFP protein and understand whether affibodies produced by a phage display platform are capable of neutralizing SARS-CoV-2 variants. Although there is a data to support some of the conclusions made in this manuscript, this reviewer would like to address several challenges:

Major comments:

The neutralization data in this study are not well supported. Single fluorescent images are provided and neutralization extrapolated without an understanding of cell density in each image or appropriate controls in some cases (see detailed comments). Without this information it is extremely difficult to validate the presence of true neutralization and the major emphasis of this manuscript.

This reviewer would appreciate additional evidence of affibody binding to endogenous respiratory cell lines that express human ACE2 to ensure that binding remains in an unmodified / biologically relevant cell line.

Detailed comments:

Single references of critical statements are inadequate to support many statements that can be validated by multiple studies at this point in the pandemic.

There is very little methodological description of the serological ELISA that was performed and would not be reproducible from this method. This should be described in much better detail.

Figure 4C: Statistical analysis of titer, area under the curve, EC_{50} etc. would help provide additional support for this conclusion.

Figure 4D-E: Serum neutralization should be validated with quantification of fluorescent intensity or by additional surrogate assays. Single images demonstrate a potential trend however this does not adequately support the conclusion that all RBMFP⁺ sera provides a neutralizing effect. In addition, the PBS group is not represented in either Figure 4D and 4E which does not provide a point of comparison and no phase contrast images providing a reference for cell number per image is provided.

Figure 5C-D: Labeling of the gels and blots would help provide additional clarity to the figure.

Figure 7: Phase contrast images should be provided and luminescent / fluorescent intensity values should be normalized to the area of cells per well. Without normalizing to cell number / area per image it is possible that a reduction in luminescent / fluorescent intensity is merely an artifact of fewer cells per image. Additionally, a statement on the number of images analyzed per sample should be provided.

Figure 7D: Statistical analysis of multiple groups should be performed using ANOVA, not *t*-test.

Lines 56-57: There is no evidence provided to suggest that the RBMFP provides neutralizing protection *in vivo* in this manuscript

Line 69: Reference number seven only refers to vaccine resistance of the delta variant. This statement should be qualified by additional studies demonstrating viral resistance to existing vaccines.

Lines 61-71: Statements in introduction should be referenced in more detail. Many studies exist supporting this logic and single references do not adequately describe the literature encompassing this topic.

Lines 73-79: References should be included to support these statements.

Lines 99-101: Further references regarding the structure, function, and utility of affibodies would be beneficial. For example: Frejd, F., Kim, KT. Affibody molecules as engineered protein drugs. *Exp Mol Med* **49**, e306 (2017). <https://doi.org/10.1038/emm.2017.35>

Lines 127-128: Is this statement supported by data in the manuscript? Neutralizing capacity *in vitro* does not necessarily translate to *in vivo*.

Line 178: What does “correct sequence” mean?

Line 225: Reference the mechanism of SARS-CoV-2 entry to cells.

Line 227: “High level of specific...”. This phrasing is not valid. No comparison for whether this degree of IgG production qualifies as high. Please revise by comparing to a control or remove “high level”.

Lines 253-254: This reviewer would appreciate clarity from the author’s as to whether there is evidence an advantage of affibodies against heavily glycosylated proteins over mAbs. If there is no evidence to support this, this statement should be revised to reflect this.

Lines 247, 257-262: The K_D of bamlanivimab is 0.071nM and blocks ACE2 spike interactions with an IC_{50} of 0.17nM. The K_D of estevimab is 6.45nM and blocks ACE2-spike interactions with an IC_{50} of 0.32nM. It would be useful to put the affinity and neutralizing capacity of the affibodies in direct context with these existing FDA-authorized drugs for discussion. The values provided in line 247 are misleading if not in the same units as the affibody molecules. K_D and IC_{50} values of affibody molecules greater than 10-fold higher than existing drugs should be discussed and statements regarding prospective efficacy tempered accordingly.

Response to reviewers' comments

Reviewer #1 (Comments for the Author):

The authors selected the domains of RBM and FP from SARS-CoV-2 S protein that binds and fuses with cell membrane receptors as targets, prepared the RBMFP fusion protein using a prokaryotic expression system, and identified its immunogenicity and antigenicity. Further, they selected the novel affibodies with phage-display technology and obtained affibodies with binding affinity to RBMFP. Then, the neutralization effect of affibodies on SARS-CoV-2 pseudovirus was evaluated.

The authors present compelling data to support the binding specificity of these affibodies and their neutralizing efficacy against SARS-CoV-2 pseudovirus. The paper is well written and interesting, especially Z14 and Z171 neutralize against Omicron variants, a dominant epidemic today, so the researchers have potential and application prospects; however, some concerns have been raised during the revision process

Response to Reviewer 1:

Here is a point-by-point response to the reviewer's comments and concerns.

Q: Please explain or evaluate affibodies compare with neutralizing antibodies against the SARS-CoV-2 virus relative to advantage. Is there any affibody being developed specifically for clinical utilization?

A: We appreciate the time and effort the reviewer has dedicated to providing valuable feedback on our manuscript. We agree with the reviewer's assessments. Compared with SARS-CoV-2 neutralizing antibodies, the affibodies have several advantages, including a smaller molecular size of 6.5 kDa and only 1/23 size than 150 kDa of antibodies. Lim et al. reported that some FDA-approved antibodies against SARS-CoV-2 delivered intravenously need several gram doses for efficient neutralization. The reason may be low efficiency across the plasma-lung barrier for pulmonary infection treatment (1). Moreover, affibodies are expressed in prokaryotic expression systems compared with most SARS-CoV-2 antibodies expressed in mammalian cells. The large-scale production of mAbs usually takes an average of 3 to 6 months, thus would be a costly and time-consuming process during the pandemic (2). Furthermore, attributing to their smaller sizes, affibodies can be easily modified and constructed for further improvement (Be modified into dimer/trimer affibody or the replacement of the N/D/A (asparagine/aspartic acid/alanine) sequence near the C segment of the affibody, and certain residue's mutation, the affinities of affibodies can be improved 100 or more times) (3).

As we know, several affibodies have been developed and registered at clinicalTrials.gov (ClinicalTrials.gov is a database of privately and publicly funded clinical studies conducted worldwide), such as ABY-025 (NCT02095210), ABY-039 (NCT03502954), and HPark2 (NCT04267900) for early phase I or phase I/II clinical trials, which have shown great potential clinical applications and prospect of these affibodies.

Thank you for this comment. As suggested, changes have been made in the revised manuscript (Lines 268-271)

Q: The authors used the RBMFP sequence origin of the SARS-CoV-2 prototype. Why not use the currently popular omicron? Are there any reasons for this?

A: We appreciate the reviewer's valuable comments. When we started our research, no delta or omicron variants were reported. During our research process, delta and omicron emerged, so we examined whether RBMFP origin from the prototype can induce antibodies in mice to neutralize the variants. As you commented, it may be better to use omicron to induce antibodies for the popular variants. Therefore, further research will compare the differences between antibodies induced by omicron, prototype, and other variants.

Q: The data in the manuscript demonstrates that the prokaryotic recombinant RBMFP protein is immunogenic, and its specific immune mouse serum also has the effect of neutralizing the pseudovirus of the new coronavirus variant. Compared with the currently marketed vaccines, how are they different?

A: Thanks for the comments. The currently marketed vaccines have been used widely and helped control the current pandemic. However, most marketed vaccines contain full-length S protein or the mRNA that translates S protein. In the virus, the spike region is extensively glycosylated, and only the RBD within the S1 subunit is well-exposed to antibody recognition (4), thus modulating the variants' escape from immune surveillance that induced by vaccines developed. In contrast, RBMFP is glycosylation free but still has RBM and other major antigen regions, such as FP and cleavage sites, which also neutralize some variants. Further, most full-length S protein vaccines and completely inactivated vaccines are prepared in eukaryotic expression systems. While the RBMFP are prepared in the prokaryotic expression system and may have advantages in future use (5).

Q: The authors point out that they have produced "high-affinity" binders. The authors report single-digit micromolar KDs calculated based on the binding kinetics using SPR. Although there is such clear cell binding and

measurable in vitro, a binder with μM affinity may not be ideal for the preventive or therapeutic agent. Whether further improvements are needed, or how to modify them further to increase their affinity for further development and application?

A: We understand the reviewer's concerns. Here, we used an SPR assay that confirmed 4 binders with nM~ μM level affinity. The binding affinity is lower than some mAbs (6). Other researchers reported that further modifications into the dimer or trimer of these affibodies could enhance their binding affinity for their target (7). In addition, certain residue sites of mutation in the developed affibodies may also further improve their affinity (3). We have added a detailed description in the revised manuscript (Lines 275-285).

Q: Molecular docking predicts the preferred orientation, affinity, and interaction of a ligand in the binding site of a protein. Could the authors explain the binding sites of Z14 and Z171 affibody molecules?

A: Thank you for the comment. As shown in Fig.8 and Table 2, Z14 may bind with RBM through interactions of **K7-I468**, **E8-N450**, **Y10-T470**, **D14-S494**, **D18-R408**, **N21-G504/Y505**, **E28-F486**. Z171 may bind with RBM through interactions of **D2-T478**, **R10/Y14-N481**, **L22/N23-N450**, **A25-S494**, **E28-Y489**, **A32-E484**. The **mark in red** are residues in **affibodies**, and the **mark in black** are residues in **RBM**. The mutated site

of Z14 compared with Zwt in the binding regions are Y10/D14/D18/E28, and in Z171 are R10/Y14/A25/E28/A32. When comparing the mutated residues in affibodies binding with RBM, Z171 has five, and Z14 has only four, which may be why Z171 has a higher affinity than Z14. Both Z14 and Z171 have E28 residue binding with F486 and Y489. That's the hydrophobic interactions contributed by F486/Y489 in the T470-F490 loop of spike protein, which is crucial for the binding (8) and may be why Z14 and Z171 still neutralize pseudoviruses of Delta and Omicron.

Q: Common English language errors, for example, Fig. 2 and 3 legends. Overall, the text is good. Just some minor corrections are needed in the manuscript.

A: We agree with the reviewer's assessment, and changes have been made accordingly in the revised manuscripts (Fig. 2 and 3 legends).

References

- (1)Lim HT, Kok BH, Lim CP, Abdul Majeed AB, Leow CY, Leow CH. 55. Biomed Eng Adv. 2022 Dec;4:100054. doi: 10.1016/j.bea.2022.100054. Epub 2022 Sep 18. PMID: 36158162; PMCID: PMC9482557.
- (2)Y. Wu, et al., Identification of human single-domain antibodies against SARSCoV-2, Cell Host Microbe 27 (6) (2020) 891–898.
- (3)Hu X, Li D, Fu Y, Zheng J, Feng Z, Cai J, Wang P. Advances in the Application of Radionuclide-Labeled HER2 Affibody for the Diagnosis and Treatment of Ovarian Cancer. Front Oncol. 2022 Jun 15;12:917439. doi: 10.3389/fonc.2022.917439. PMID: 35785201; PMCID: PMC9240272.
- (4)Fernández A. SARS-CoV-2 Glycosylation Suggests That Vaccines Should Have Adopted the S1 Subunit as Antigen. ACS Pharmacol Transl Sci. 2021 Feb 5;4(2):1016-1017. doi: 10.1021/acspsci.1c00036. PMID: 33860218; PMCID: PMC8033745.

- (5) Ke Q, Sun P, Wang T, Mi T, Xu H, Wu J, Liu B. 2022. Non-glycosylated SARS-CoV-2 RBD elicited a robust neutralizing antibody response in mice. *J Immunol Methods* 506:113279
- (6) Hwang YC, Lu RM, Su SC, Chiang PY, Ko SH, Ke FY, Liang KH, Hsieh TY, Wu HC. 2022. Monoclonal antibodies for COVID-19 therapy and SARS-CoV-2 detection. *J Biomed Sci* 29:1.
- (7) Shan L. 18F-Labeled N-(4-fluorobenzylidene)oxime-dimeric (ZHER2:477)2. 2011 Nov 2 [updated 2012 Jan 4]. In: *Molecular Imaging and Contrast Agent Database (MICAD)* [Internet]. Bethesda (MD): National Center for Biotechnology Information (US); 2004–2013. PMID: 22238798.
- (8) Chen H, Kang Y, Duan M, Hou T. Regulation Mechanism for the Binding between the SARS-CoV-2 Spike Protein and Host Angiotensin-Converting Enzyme II. *J Phys Chem Lett.* 2021 Jul 15;12(27):6252-6261. doi: 10.1021/acs.jpcclett.1c01548. Epub 2021 Jul 1. PMID: 34196550; PMCID: PMC8265532.

Reviewer #2

Overview: This manuscript provides an interesting development narrative of four novel affibodies targeting the RBMFP of the SARS-CoV-2 spike protein. Effort was taken to evaluate the antigenicity of the RBMFP protein and understand whether affibodies produced by a phage display platform are capable of neutralizing SARS-CoV-2 variants. Although there is a data to support some of the conclusions made in this manuscript, this reviewer would like to address several challenges:

Response to Review2:

We feel great thanks for your professional review work on our article. As you are concerned, several challenges need to be addressed. According to your suggestions, we have made extensive corrections to our previous draft, and the detailed corrections are listed below.

Major comments:

Q: The neutralization data in this study are not well supported. Single fluorescent images are provided and neutralization extrapolated without an understanding of cell density in each image or appropriate controls in some cases (see detailed comments). Without this information it is extremely difficult to validate the presence of true neutralization and the major emphasis of this manuscript.

A: We appreciate the time and effort the reviewer dedicated to providing valuable feedback on our manuscript, and we completely agree with the reviewer's assessment. According to the reviewer comments, single fluorescent images with under fluorescences and white light showed the cell density of each image was added in the revised manuscript (Figure 4D, E); the cell numbers were counted through Image Pro Plus 6.0 software and, also added (Fig. S2 B).

Figure 4

D

E

Fig. S2

Q: This reviewer would appreciate additional evidence of affibody binding to endogenous respiratory cell lines that express human ACE2 to ensure that binding remains in an unmodified / biologically relevant cell line.

A: Thank you for the comments. Our laboratory ever used several different cell lines expressing human ACE2 including Vero-E6 (The cell line used for preparing SARS-CoV-2 inactivated vaccine), Huh-7, HUVEC, HEK-293T, HEK-293T-ACE2, to investigate for pseudovirus infection. HEK-293T-ACE2 is the only cell that showed ideal efficiency in pseudo-

virus infection results (FigA B, data in another manuscript and not published). Based on the above and from other previously published papers (1-3), we selected HEK-293T-ACE2 cells for scientific research. However, according to your suggestion, we will further investigate the endogenous respiratory cell lines that express human ACE2, such as Calu-3 or other cell lines, to conduct the following research.

Fig A: GFP expressed in different cells infected with various concentrations of prototype SARS-CoV-2 pseudovirus. Fig. B: Luminescent signals detected in different cells infected with various concentrations of prototype SARS-CoV-2 pseudovirus.

References:

- (1)Zhou YQ, Wang K, Wang XY, Cui HY, Zhao Y, Zhu P, Chen ZN. SARS-CoV-2 pseudovirus enters the host cells through spike protein-CD147 in an Arf6-dependent manner. *Emerg Microbes Infect.* 2022 Dec;11(1):1135-1144. doi: 10.1080/22221751.2022.2059403. PMID: 35343395; PMCID: PMC9037224.
- (2)Freedenberg AT, Pan CH, Diehl WE, Romeiser JL, Hwang GR, Leiton CV, Muecksch F, Shroyer KR, Bennett-Guerrero E. Neutralizing activity to SARS-CoV-2 of convalescent and control plasma used in a randomized

controlled trial. *Transfusion*. 2021 May;61(5):1363-1369. doi: 10.1111/trf.16283. Epub 2021 Feb 19. PMID: 33448402; PMCID: PMC8014203.

(3) Rao L, Xia S, Xu W, Tian R, Yu G, Gu C, Pan P, Meng QF, Cai X, Qu D, Lu L, Xie Y, Jiang S, Chen X. De-coy nanoparticles protect against COVID-19 by concurrently adsorbing viruses and inflammatory cytokines. *Proc Natl Acad Sci U S A*. 2020 Nov 3;117(44):27141-27147. doi: 10.1073/pnas.2014352117. Epub 2020 Oct 6. PMID: 33024017; PMCID: PMC7959535.

Detailed comments:

Q: Comment1: Single references of critical statements are inadequate to support many statements that can be validated by multiple studies at this point in the pandemic.

A: Thank you for the suggestion. According to your comment, we have added more references to support these statements in the revised manuscript (Line 72, ref 8-10; lines 77-79, ref 12-14; lines 89-91, ref 20-21 and lines 99-103, ref 7, 24-28).

References revised

(8)Servellita, V., et al., Neutralizing immunity in vaccine breakthrough infections from the SARS-CoV-2 Omicron and Delta variants. *Cell*, 2022. 185(9): p. 1539-1548.e5.

(9)Ren, S.Y., et al., Omicron variant (B.1.1.529) of SARS-CoV-2: Mutation, infectivity, transmission, and vaccine resistance. *World J Clin Cases*, 2022. 10(1): p. 1-11.

(10)Tuekprakhon, A., et al., Antibody escape of SARS-CoV-2 Omicron BA.4 and BA.5 from vaccine and BA.1 serum. *Cell*, 2022. 185(14): p. 2422-2433.e13.

(12)Velusamy, P., et al., SARS-CoV-2 spike protein: Site-specific breakpoints for the development of COVID-19 vaccines. *J King Saud Univ Sci*, 2021. 33(8): p. 101648.

(13)Salvatori, G., et al., SARS-CoV-2 spike protein: An optimal immunological target for vaccines. *Journal of Translational Medicine*, 2021. 1(1).

(14)Yao, H., et al., Molecular Architecture of the SARS-CoV-2 Virus. *Cell*, 2020. 183(3): p. 730-738.e13.

(20)Johnson, B.A., et al., Loss of furin cleavage site attenuates SARS-CoV-2 pathogenesis. *Nature*, 2021. 591(7849): p. 293-299.

(21)Chan, Y.A. and S.H. Zhan, The Emergence of the Spike Furin Cleavage Site in SARS-CoV-2. *Mol Biol Evol*, 2022. 39(1).

- (24) VanBlargan, L.A., et al., An infectious SARS-CoV-2 B.1.1.529 Omicron virus escapes neutralization by therapeutic monoclonal antibodies. *Nat Med*, 2022. 28(3): p. 490-495.
- (25) Ai, J., et al., Antibody evasion of SARS-CoV-2 Omicron BA.1, BA.1.1, BA.2, and BA.3 sub-lineages. *Cell Host Microbe*, 2022. 30(8): p. 1077-1083.e4.
- (26) Grant, O.C., et al., Analysis of the SARS-CoV-2 spike protein glycan shield reveals implications for immune recognition. *Sci Rep*, 2020. 10(1): p. 14991.
- (27) Tuccori, M., et al., Anti-SARS-CoV-2 neutralizing monoclonal antibodies: clinical pipeline. *MAbs*, 2020. 12(1): p. 1854149.
- (28) Gong, Y., et al., The glycosylation in SARS-CoV-2 and its receptor ACE2. *Signal Transduct Target Ther*, 2021. 6(1): p. 396.

Q: There is very little methodological description of the serological ELISA that was performed and would not be reproducible from this method. This should be described in much better detail.

A: Thank you for the valuable suggestion. We have made changes to the detailed description of the serological ELISA method in the revised manuscript accordingly (Method- Mice serological ELISA, lines 367-383).

Q: Figure 4C: Statistical analysis of titer, area under the curve, EC50 etc. would help provide additional support for this conclusion.

A: Thank you for your comment. The statistical analysis of titer has been added to the data, and changes have been made to Figure 4 C in the revised manuscript.

Q: Figure 4D-E: Serum neutralization should be validated with quantification of fluorescent intensity or by additional surrogate assays. Single images demonstrate a potential trend however this does not adequately support the conclusion that all RBMFP+ sera provides a neutralizing effect.

A: Thank you for the valuable comment and suggestion. The evaluation of neutralizing effect of RBMFP+Sera was validated by quantification of luciferase assay and fluorescent images with cell density (Figure 4 D-E, the white light of merged images was provided). Quantized figures were added as supplementary materials (Fig. S2), and we have properly discussed this statement in the methodology and results from sections in the revised manuscript accordingly (Lines 177-182, 384-387, 397-408).

Q: In addition, the PBS group is not represented in either Figure 4D and 4E which does not provide a point of comparison and no phase contrast images providing a reference for cell number per image is provided.

A: Thank you for the comments. In this study, a culture medium was used to dilute mouse serum, and no mice serum added group was set as a negative control, which we represented as serum-free in the original manuscript. Furthermore, we also provided white light images in the re-

vised manuscript (Figure 4 D, E), and sentences have been modified accordingly (Lines 384-387, 400-408).

Q: Figure 5C-D: Labeling of the gels and blots would help provide additional clarity to the figure.

A: Thank you for the suggestions. Changes have been made accordingly in the revised manuscript (Figure 5C-D, Legend).

Q: Figure 7: Phase contrast images should be provided and luminescent / fluorescent intensity values should be normalized to the area of cells per well. Without normalizing to cell number / area per image it is possible that a reduction in luminescent / fluorescent intensity is merely an artifact of fewer cells per image. Additionally, a statement on the number of images analyzed per sample should be provided. Statistical analysis of multiple groups should be performed using ANOVA, not *t*-test.

A: Thank you for your suggestions. As suggested, phase contrast images have been provided, cell numbers have been counted (Figure S7), and statistical analysis was performed using ANOVA (Line 505).

Q: Lines 56-57: There is no evidence provided to suggest that the RBMFP provides neutralizing protection *in vivo* in this manuscript

A: Thanks for your valuable comment. We have revised the statements that RBMFP provides neutralizing effects against pseudovirus *in vitro* (Lines 56-57).

Q: Line 69: Reference number seven only refers to vaccine resistance of the delta variant. This statement should be qualified by additional studies demonstrating viral resistance to existing vaccines. Lines 61-71: Statements in introduction should be referenced in more detail. Many studies exist supporting this logic and single references do not adequately describe the literature encompassing this topic.

A: We appreciate the reviewer's suggestions. Changes have been made accordingly in the revised manuscript (Lines 71-73).

References revised:

(7)Liu, C., et al., Reduced neutralization of SARS-CoV-2 B.1.617 by vaccine and convalescent serum. *Cell*, 2021. 184(16): p. 4220-4236.e13.

(8)Servellita, V., et al., Neutralizing immunity in vaccine breakthrough infections from the SARS-CoV-2 Omicron and Delta variants. *Cell*, 2022. 185(9): p. 1539-1548.e5.

(9)Ren, S.Y., et al., Omicron variant (B.1.1.529) of SARS-CoV-2: Mutation, infectivity, transmission, and vaccine resistance. *World J Clin Cases*, 2022. 10(1): p. 1-11.

(10)Tuekprakhon, A., et al., Antibody escape of SARS-CoV-2 Omicron BA.4 and BA.5 from vaccine and BA.1 serum. *Cell*, 2022. 185(14): p. 2422-2433.e13.

Q: Lines 73-79: References should be included to support these statements.

A: Thanks for your valuable suggestion. Changes have been made accordingly in the revised manuscript (Lines 75-82, references 11-16).

References revised:

- (11) Wu, C.R., et al., Structure genomics of SARS-CoV-2 and its Omicron variant: drug design templates for COVID-19. *Acta Pharmacol Sin*, 2022.
- (12) Velusamy, P., et al., SARS-CoV-2 spike protein: Site-specific breakpoints for the development of COVID-19 vaccines. *J King Saud Univ Sci*, 2021. 33(8): p. 101648.
- (13) Salvatori, G., et al., SARS-CoV-2 spike protein: An optimal immunological target for vaccines. *Journal of Translational Medicine*, 2021. 1(1).
- (14) Yao, H., et al., Molecular Architecture of the SARS-CoV-2 Virus. *Cell*, 2020. 183(3): p. 730-738.e13.
- (15) Ou, X., et al., Characterization of spike glycoprotein of SARS-CoV-2 on virus entry and its immune cross-reactivity with SARS-CoV. *Nat Commun*, 2020. 11(1): p. 1620.
- (16) Shang, J., et al., Cell entry mechanisms of SARS-CoV-2. *Proc Natl Acad Sci U S A*, 2020. 117(21): p. 11727-11734.

Q: Lines 99-101: Further references regarding the structure, function, and utility of affibodies would be beneficial. For example: Frejd, F., Kim, KT. Affibody molecules as engineered protein drugs. *Exp Mol Med* **49**, e306 (2017). <https://doi.org/10.1038/emm.2017.35>

A: We are grateful for this suggestion. Relevant references have been added in more detail, and statements have been revised as you suggested. Lines 105-109, structure (References added 29), function (reference added 30.31), utility (References added 32).

References revised:

(29) Chopra A. 2004. (111)In-Labeled anti-epidermal growth factor receptor Affibody PEP09239, Molecular Imaging and Contrast Agent Database (MICAD). National Center for Biotechnology Information (US), Bethesda (MD).

(30) Fu, R., et al., Antibody Fragment and Affibody ImmunPET Imaging Agents: Radiolabelling Strategies and Applications. ChemMedChem, 2018. 13(23): p. 2466-2478.

(31)Tolmachev, V. and A. Orlova, Affibody Molecules as Targeting Vectors for PET Imaging. Cancers (Basel), 2020. 12(3).

(32)Frejd, F.Y. and K.T. Kim, Affibody molecules as engineered protein drugs. Exp Mol Med, 2017. 49(3): p. e306.

Q: Lines 127-128: Is this statement supported by data in the manuscript?

Neutralizing capacity *in vitro* does not necessarily translate to *in vivo*.

A: We do agree with the reviewer. The statements have been revised in the manuscript as "Herein, we provide the first evidence that the affibody molecules are novel agents against pseudovirus of SARS-COV-2 variants" (Lines 137-139).

Q: Line 178: What does "correct sequence" mean?

A: The "correct sequence" means DNA sequences of screened affibody molecules that encoded the full length of protein containing 13 mutation residues and no premature termination. That is to say, among the 58 amino acids constituting the full length of the affibody, 13 certain mutation sites were mutated compared with Z_{WT}, and the DNA sequences in skeleton regions maintained the correct codon encoding amino acids. Revised in Line 190-191.

Q: Line 225: Reference the mechanism of SARS-CoV-2 entry to cells.

A: Thank you for your suggestion. The mechanism of SARS-CoV-2 entry to cells has been revised in the introduction and discussion (Lines 75-91, references revised 10-22, line 237-238, references revised 44-45).

References revised:

(44)Carvalho, P.P.D. and N.A. Alves, Featuring ACE2 binding SARS-CoV and SARS-CoV-2 through a conserved evolutionary pattern of amino acid residues. *J Biomol Struct Dyn*, 2021: p. 1-10.

(45) Jackson, C.B., et al., Mechanisms of SARS-CoV-2 entry into cells. *Nat Rev Mol Cell Biol*, 2022. 23(1): p. 3-20.

Q: Line 227: "High level of specific...". This phrasing is not valid. No comparison for whether this degree of IgG production qualifies as high. Please revise by comparing to a control or remove "high level".

A: We totally agree with the reviewer's assessment, and changes have been made accordingly in the revised manuscript (Line 240).

Q: Lines 253-254: This reviewer would appreciate clarity from the author's as to whether there is evidence an advantage of affibodies against heavily glycosylated proteins over mAbs. If there is no evidence to support this, this statement should be revised to reflect this.

A: We agree with the reviewer's comment. As you mentioned, we have checked whether there is evidence of an advantage of affibodies against

heavily glycosylated proteins over mAbs. However, there is no experimental evidence but only the description of the possibility to support this. Therefore, the statement has been revised accordingly (Lines 265-271). Further, your valuable comment provides us with a novel research idea.

Q: Lines 247, 257-262: The KD of bamlanivimab is 0.071nM and blocks ACE2 spike interactions with an IC50 of 0.17nM. The KD of estevimab is 6.45nM and blocks ACE2-spike interactions with an IC50 of 0.32nM. It would be useful to put the affinity and neutralizing capacity of the affibodies in direct context with these existing FDA-authorized drugs for discussion. The values provided in line 247 are misleading if not in the same units as the affibody molecules. KD and IC50 values of affibody molecules greater than 10-fold higher than existing drugs should be discussed and statements regarding prospective efficacy tempered accordingly

A: We greatly appreciate the reviewer's constructive comments, and the revised manuscript's changes have been made accordingly. The values in line 247 in the original text have been modified in the same units nmol/L as affibody molecules (Line 261). The discussion of affinity and neutralizing capacity of the affibodies with FDA-authorized drugs have been revised (Lines 275-285)

November 21, 2022

Prof. Lifang Zhang
Wenzhou Medical University
306 hualongqiao road
Wenzhou, Zhejiang province
China

Re: Spectrum03562-22R1 (Novel Affibody Molecules Specifically Bind to SARS-CoV-2 Spike Protein and Efficiently Neutralize Delta and Omicron Variants)

Dear Prof. Lifang Zhang:

Your manuscript has been accepted, and I am forwarding it to the ASM Journals Department for publication. You will be notified when your proofs are ready to be viewed.

Sincerely,

Yongjun Sui
Editor, Microbiology Spectrum
